**Data Availability Statement:** The cryo-EM density maps and structures for FMDV-AWH-W125 and FMDV-AWH-W2 have been deposited at the Electron Microscopy Data Bank (EMDB) and the

# Conserved antigen structures and antibody-driven variations on foot-and-mouth disease virus serotype A revealed by bovine neutralizing monoclonal antibodies

**Kun Li[1☯], Yong He[2,3☯], Li Wang[1], Pinghua Li[1], Huifang Bao[1], Shulun Huang[1], Shasha Zhou[1], Guoqiang Zhu[1], Yali Song[1], Ying Li[1], Sheng Wang[1], Qianliang Zhang[1], Pu Sun[1], Xingwen Bai[1], Zhixun Zhao[1], Zhiyong Lou[3]\*, Yimei Cao[1]\*, Zengjun Lu[1]\*, Zaixin Liu[1]**

**1** State Key Laboratory for Animal Disease Control and Prevention, College of Veterinary Medicine, Lanzhou University, National Foot-and-Mouth Diseases Reference Laboratory, Lanzhou Veterinary Research Institute, Chinese Academy of Agricultural Sciences, Lanzhou (P.R. China), **2** College of Pharmaceutical Sciences, Shandong University, Jinan, China, **3** MOE Key Laboratory of Protein Science & Collaborative Innovation Center of Biotherapy, School of Medicine, Tsinghua University, Beijing, China

☯ These authors contributed equally to this work.
\* louzy@mail.tsinghua.edu.cn (ZL); caoyimei@caas.cn (YC); luzengjun@caas.cn (ZL)

## Abstract

Foot-and-mouth disease virus (FMDV) serotype A is antigenically most variable within serotypes. The structures of conserved and variable antigenic sites were not well resolved. Here, a historical A/AF72 strain from A22 lineage and a latest A/GDMM/2013 strain from G2 genotype of Sea97 lineage were respectively used as bait antigen to screen single B cell antibodies from bovine sequentially vaccinated with A/WH/CHA/09 (G1 genotype of Sea97 lineage), A/GDMM/2013 and A/AF72 antigens. Total of 39 strain-specific and 5 broad neutralizing antibodies (bnAbs) were isolated and characterized. Two conserved antigenic sites were revealed by the Cryo-EM structures of FMDV serotype A with two bnAbs W2 and W125. The contact sites with both VH and VL of W125 were closely around icosahedral threefold axis and covered the B-C, E-F, and H-I loops on VP2 and the B-B knob and H-I loop on VP3; while contact sites with only VH of W2 concentrated on B-B knob, B-C and E-F loops on VP3 scattering around the three-fold axis of viral particle. Additional highly conserved epitopes also involved key residues of $_{VP1}58$, $_{VP1}147$ and both $_{VP2}72$ / $_{VP1}147$ as determined respectively by bnAb W153, W145 and W151-resistant mutants. Furthermore, the epitopes recognized by 20 strain-specific neutralization antibodies involved the key residues located on VP3 68 for A/AF72 (11/20) and VP3 175 position for A/GDMM/2013 (9/19), respectively, which revealed antigenic variation between different strains of serotype A. Analysis of antibody-driven variations on capsid of two virus strains showed a relatively stable VP2 and more variable VP3 and VP1. This study provided important information on conserve and variable antigen structures to design broad-spectrum molecular vaccine against FMDV serotype A.

Protein Data Bank (PDB) under the following accession numbers with corresponding link: EMD-34213 (https://www.ebi.ac.uk/emdb/EMD-34213) and 8GRR (https://doi.org/10.2210/pdb8GRR/pdb) for FMDV-AWH-W125; EMD-34238 (https://www.ebi.ac.uk/emdb/EMD-34238) and 8GSP (https://doi.org/10.2210/pdb8GSP/pdb) for FMDV-AWH-W2. The sequences of all the bovine-derived monoclonal antibodies have been deposited in GenBank (Accession Numbers OR540314 to OR540401).

**Funding:** This work was supported by grants from the National Key R&D Program of China (2021YFD1800304 to Z.Lu.), the National Natural Science Foundation of China (Nos. 31902288 to K. L. and 32072873 to Y.Cao.) and China Postdoctoral Science Foundation (2023M732084 to Y.H.). The funders had no role in study design, data collection and analysis, decision to publish, or preparation of the manuscript.

**Competing interests:** The authors have declared that no competing interests exist.

## Author summary

Bovine is susceptible host to foot-and-mouth disease virus (FMDV) and neutralization antibodies provide vital protection in defending viral infection, concurrently driving viral evolution in host. Herein, using single B cell antibody technology, we isolated and characterized a panel of 44 bovine-derived neutralizing monoclonal antibodies against FMDV serotype A, including 39 strain-specific and 5 broad neutralizing antibodies (bnAbs) against both A22 and Sea97 lineages representative strains. We revealed at least four conserved antigen sites including two sites on VP1 and each one on VP2/VP3, which exist on viral capsid surface and can induce bnAb response to FMDV serotype A in vivo. Additionally, antibody-driven variations showed shrinkage and appearance of strain-specific antigen epitopes were found on VP3 68 and 175 positions of FMDV serotype A. To sum up, this study provided conserved antigen structures and strain-specific epitopes information to guide the design of broad vaccine molecular against FMDV serotype A.

## Introduction

Foot-and-mouth disease virus (FMDV) is an extremely contagious pathogen affecting bovine, sheep, swine and other cloven-hoofed animals, and globally exists as seven serotypes: A, O, C, Asia1, and SAT1-3. Serotype C FMDV has not been reported since 2004 and it is now considered to be extinct. Of which, serotype A was widely distributed in the Asia, Africa and Euro-SA regions, and considered to be the most variable and antigenic diversity with 26 genotypes, making obtain of broad antigen-spectrum vaccine difficult [1]. Neutralizing antibodies (NAbs) provided vital protection against FMDV and concurrently driven viral evolution under immune pressure in bovine [2–5]. In-depth study of the antigenic structure of FMDV can guide the molecular design of broad vaccine and facilitate the discovery of candidate vaccine strains [6,7]. Monoclonal NAbs were also crucial tools for dissecting of antigenic structure of FMDV and revealing the viral variant under immune pressure.

Bovine was susceptible host to FMDV, and bovine-derived NAb could be more precise tools to discover the viral antigenic structure reflecting the immune response in vivo. Indeed, using bovine broad neutralizing monoclonal antibody (bnAb), we have identified a novel cross-serotype antigen site and revealed the conserved antigen structure between serotype O and A [8]. We also identified two cross-protective antigen sites on serotype O by bovine intra-type bnAb and firstly revealed a vulnerable site of viral particle [9]. The present work confirmed that bovine-derived antibodies benefit to identify the protective as well as latent determinations on FMDV capsids. For serotype A, the identification of antigen sites was mainly based on murine-derived monoclonal antibodies (mAbs) [10–14]. On A5 strain, two antigen sites were earliest reported involving the amino acids (AA) 198 on C-terminus of VP1 and the 72, 79 AA on B-C loop of VP2 defined by neutralization-resistant mutants [10]. On A10 strain, three groups of antigenic sites were identified, including the G-H loop and C-terminus of VP1, the 58–70, 139 and 195 AA on VP3 and the 80 AA on VP2. Besides, the antigenic sites on VP3 were targeted by most of NAbs [11,12]. Unlike A5 and A10 strains, the G-H loop on VP1 seems a major antigenic site on A12, A22 and A24 strains [13–15], and the key determinations involving this site are diverse, consisting of both conformational and linear epitopes [16,17]. Obviously, the antigenic structure of divergent serotype A reported above was only focused on some strains of specific lineage, and the conserved antigenic structure and viral evolution of FMDV serotype A were less understood.

There are three representative isolates (A/AF72, A/WH/CHA/09 and A/GDMM/2013) of FMDV serotype A in China, which may represent antigenic diversity or antigenic evolutionary history of ASIA topotype of serotype A viruses. The A/AF72 was a historical strain derived in diseased bovine in 1972 from Feicheng, Shandong province in Eastern China belonging to the A22 lineage [18]. In 2009, the A/WH/CHA/09 strain was first isolated from bovine in 2009 in Wuhan, Hubei province in central China, which belongs to the G1 genotype of Southeast Asia 97 (Sea97) lineage involving several clinical cases from 2009 to 2010 [19]. The latest A/GDMM/2013 strain was first reported in 2013 in Maoming, Guangdong province in southern China, which is classified into the G2 genotype of Sea97 lineage [20]. Both A/WH/CHA/09 and A/GDMM/2013 strain were introduced from neighbor countries of Southeast Asia. These strains tend to infect bovine, and although A/GDMM/2013 was initially isolated in swine, it mainly causes disease in bovine. The historical strain A/AF72 was closed to the prototype strain ($A_{22}$/IRQ/24/64) for ASIA topotype with 90.1% amino acid homology on VP1, but distant from the A/GDMM/2013 and A/WH/CHA/09 strains, showing only 86.8% and 87.3% of VP1 amino acid homology. A/AF72 and A/WH/CHA/09 strains were both vaccine strains used in different time and showed good antigenicity. The antigenic diversity of the three strains isolated at different periods could be considered as the good resources for investigating the antigenic evolution for serotype A.

Therefore, in this study, we chose natural host bovine and used the historical A/AF72 and the latest A/GDMM/2013 respectively as bait antigens to isolate FMDV serotype A specific NAbs via single B cells antibody technique from peripheral blood mononuclear cells (PBMCs) of bovine sequentially vaccinated with A/WH/CHA/09, A/GDMM/2013 and A/AF72 antigens. The panels of FMDV serotype A broad and strain-specific NAbs of bovine origin were used to reveal the conserved and variable antigenic sites, which will provide more precise structural insight to explain the antigenic variation of different FMDV type A isolates.

## Results

### Production of bovine broad and strain-specific neutralizing antibodies against FMDV serotype A

To evaluate the width of antigen spectrum of different FMDV serotype A strains, we chose the 146S particle of the A/AF72 strain and A/GDMM/2013 strain (S1 Fig) respectively as bait antigen to isolate antigen-specific single B cells from PBMCs of bovine. As revealed in the flow cytometry, we first excluded the IgM$^+$ naïve B cells and sorted the antigen-specific class-switched B cell to obtain high affinity antibody (Fig 1A and 1C). The FMDV serotype A-binding class-switched B cells were a scarce population and existed in both CD21$^+$IgM$^-$ and CD21$^-$IgM$^-$ populations (Fig 1D and 1E). The proportion of B cells binding to A/AF72 was comparable to that of B cells binding to A/GDMM/2013, accounting approximately 0.05% of bovine total PBMCs (Fig 1F and 1G). The equal portion of antigen-binding B cells were separately sorted and used to produce bovine mAbs following the protocol in our previous description via the FACS-based single B cell antibody technique [21]. We finally successfully obtained a total of 44 FMDV serotype A-neutralizing mAbs that were consisted of 25 mAbs isolated from A/AF72 antigen-binding B cells (Table 1) and 19 mAbs from A/GDMM/2013 antigen-binding B cells (Table 2). The binding characteristics of these mAbs were evaluated using indirect enzyme-linked immunosorbent assay (ELISA) and the results showed that most of these mAbs bound with both the intact (146S) particle and dissociated (12S) pentamer (S2 Fig). Some mAbs recognized viral particle in a strictly 146S-dependent way and the 146S-specific binding was also revealed in other strains of FMDV serotype A [22]. To verify the viral neutralization titer and width, these bovine mAbs were respectively tested against three strains of

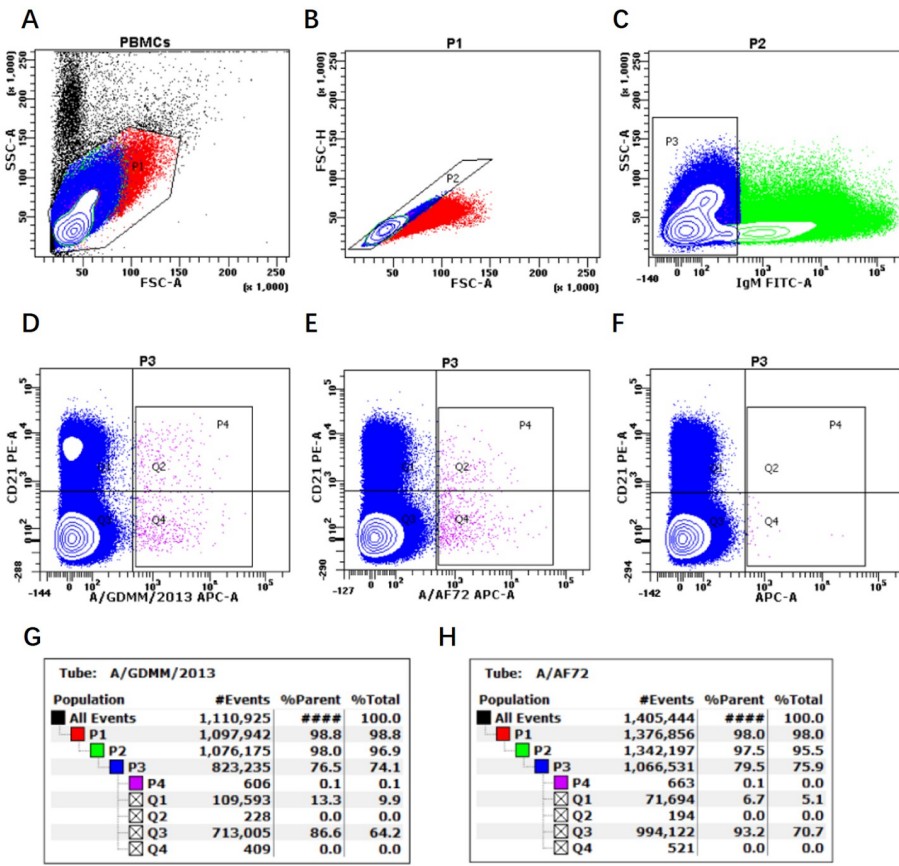

**Fig 1. Sorting and proportion of FMDV serotype A-specific B cells via different bait antigens by flow cytometry.**
Bovine PBMCs (A) were analyzed by 3-color flow cytometry and the gate P1 (B) was selected to exclude cells debris with lower values of SSC-A and FSC-A, and further analyzed singlets in gate P2 (C) based on diagonal streak of the FSC-A and FSC-H plot. The class-switched (IgM⁻) B cells in gate P3 were used to check the distribution of CD21 and FMDV-specific cells, in the presence of FMDV bait antigen A/GDMM/2013 (D) and A/AF72 (E), respectively. The demarcation line between FMDV-specific binding and nonbinding B cells was set according to the FMO control (F) in absence of bait antigen. The CD21$^{+/-}$ Antigen$^+$ B cells in gate P4 were separately sorted for preparation of bovine mAbs. Appropriately one million bovine PBMCs were collected to analyze the distribution proportion of A/GDMM/2013-binding (G) and A/AF72-binding (H) B cells in circulating peripheral blood.

serotype A viruses, A/AF72 strain of A22 lineage, A/WH/CHA/09 and A/GDMM/2013 strains of Sea97 lineage, using virus neutralization test (VNT) on BHK-21. As shown in Table 1, the most of the A/AF72 antigen-derived mAbs (20/25) showed broad neutralizing activity against two or three representative strains. Of which, five mAbs (W2, W125, W151, W145 and W153) showed potently neutralizing ability (VN titer <25 µg/mL) against three representative strains from two distinct lineages, depicting the bnAbs feature against FMDV serotype A. In contrast, the neutralizing mAbs derived from the A/GDMM/2013 antigen showed less neutralizing width and most of NAbs (16/19) were strain-specific NAbs against itself and only three (3/19) neutralizing both A/WH/CHA/09 and A/GDMM/2013 in the Sea97 lineage. However, none of the A/GDMM/2013 derived NAbs in Table 2 could neutralize the A/AF72 strain in A22 lineage. Thus, contrastive analysis of the neutralizing width of all the obtained mAbs suggested the difference in antigen spectrum between the A/AF72 and A/GDMM/2013, and the A/AF72 might have a broader antigen coverage and benefit to induce bnAb response against FMDV serotype A in bovine.

**Table 1. Neutralization titer and width of bovine-derived mAbs which were obtained by single B cell antibody technique using the A/AF72 strain as bait antigen.**

| | | FMDV serotype A | | |
|---|---|---|---|---|
| No | Clone | A/AF72 | A/WH/CHA/09 | A/GDMM/2013 |
| 1 | W2 | 7.78 | 5.47 | 2.73 |
| 2 | W125 | 4.14 | 2.07 | 16.56 |
| 3 | W145 | 3.91 | 15.63 | 7.81 |
| 4 | W151 | 24.38 | 24.38 | 12.19 |
| 5 | W153 | 3.05 | 6.10 | 3.05 |
| 6 | W73 | 21.88 | 10.94 | - |
| 7 | W3 | 1.17 | 9.38 | - |
| 8 | W7 | 6.25 | 25.00 | - |
| 9 | W18 | 7.50 | 7.50 | - |
| 10 | W49 | 0.98 | 1.95 | - |
| 11 | W68 | 3.30 | 3.30 | - |
| 12 | W72 | 4.83 | 4.83 | - |
| 13 | W92 | 6.25 | 6.25 | - |
| 14 | W93 | 4.37 | 8.75 | - |
| 15 | W104 | 5.16 | 5.16 | - |
| 16 | W118 | 3.13 | 10.31 | - |
| 17 | W121 | 10.00 | 20.00 | |
| 18 | W124 | 1.02 | 4.06 | - |
| 19 | W160 | 1.77 | 14.1 | |
| 20 | W178 | 50.00 | 6.25 | |
| 21 | W185 | 15.63 | 15.63 | |
| 22 | W66 | 4.45 | - | - |
| 23 | W99 | 4.06 | - | - |
| 24 | W140 | 5.00 | - | - |
| 25 | W155 | 2.36 | - | - |

Values are virus neutralization (VN) titer in μg/ml. An VN value in 50 μg/ml was used as a cut-off for neutralization and >50 μg/ml was determined as no virus neutralizing activity.

## Overall architecture of the Cryo-EM complexes FMDV-AWH-W125 and FMDV-AWH-W2

Less information about conservative antigen structure on the VP2 and VP3 of FMDV serotype A was available. Neutralization escape mutants suggested the recognition of bnAbs W125 and W2 was involved in VP2 and VP3. The two bnAbs can effectively neutralize all the three serotype A strains. Contrastively, A/WH/CHA/09 strain had been evaluated to have good immunogenicity and particle stability, and currently used as a vaccine strain for preventing serotype A viruses. Thus, we determined the structures of A/WH/CHA/09 in complex with W125 scFv (FMDV-AWH-W125) and W2 scFv (FMDV-AWH-W2), respectively. The acquired cryo-EM particle images clearly indicated that the scFv attached to the surface of the virions (S3 Fig). The cryo-EM reconstruction showed that three-molecule W125 scFv and W2 scFv were both bound to the FMDV-AWH capsid around icosahedral threefold axis, but with W125 positioned closer to the center of threefold axis compared to W2 scattering around it (Fig 2). A total of 60 copies scFv were bound to each mature virion. The final resolution of the cryo-EM reconstruction was estimated by the gold standard Fourier shell correlation (FSC) = 0.143

**Table 2. Neutralization titer and width of bovine-derived mAbs using the A/GDMM/2013 as bait antigen.**

| No | Clone | FMDV serotype A | | |
| | | A/AF72 | A/WH/CHA/09 | A/GDMM/2013 |
|---|---|---|---|---|
| 1 | R5 | - | 3.20 | 2.27 |
| 2 | R95 | - | 3.50 | 2.50 |
| 3 | R118 | - | 3.00 | 3.00 |
| 4 | R153 | - | - | 0.90 |
| 5 | R161 | - | - | 2.06 |
| 6 | R164 | - | - | 3.60 |
| 7 | R183 | - | - | 4.69 |
| 8 | R136 | - | - | 2.36 |
| 9 | R135 | - | - | 17.76 |
| 10 | R109 | - | - | 9.07 |
| 11 | R104 | - | - | 2.00 |
| 12 | I56 | - | - | 1.25 |
| 13 | R53 | - | - | 5.56 |
| 14 | R63 | - | - | 23.89 |
| 15 | R65 | - | - | 16.00 |
| 16 | R121 | - | - | 46.58 |
| 17 | I22 | - | - | 28.13 |
| 18 | R125 | - | - | 7.50 |
| 19 | R127 | - | - | 8.88 |

Values are virus neutralization (VN) titer in µg/ml. An VN value in 50 µg/ml was used as a cut-off for neutralization and >50 µg/ml was determined as no virus neutralizing activity.

criterion to be 3.75 Å for the FMDV-AWH-W2 complex and 3.72 Å for the FMDV-AWH-W125 complex (S3 Fig). In both cases, the cryo-EM densities were of sufficient quality to allow for atomic modeling of most of the FMDV capsid proteins and the variable loops of the scFv antibodies that are responsible for virus recognition (S4 Fig).

## The conserved antigenic sites on VP2 and VP3 of FMDV serotype A resolved by Cryo-EM complex structures

To obtain information on the epitopes of W2 and W125, the scFv molecular interactions were analyzed using CCP4 software, and the footprints were defined by the atoms in the virus that were closer than 4 Å to any atom in the bound scFv molecule by using RIVEM computer program. As revealed in FMDV-AWH-W125 complex (Fig 3), W125 contacts with the βB and BC/EF/HI-loops of VP2, the B-B knob and HI-loop of VP3 within one protomer. Residues in VP2 βB ($_{VP2}$D68), BC-loop ($_{VP2}$T70, $_{VP2}$T71, $_{VP2}$K73 and $_{VP2}$H77), EF-loop ($_{VP2}$E131 and $_{VP2}$K137) and HI-loop ($_{VP2}$Q196) interact with residues in heavy chain complementarity-determining region 3 (HCDR3) ($_{VH}$Y117, $_{VH}$R120, $_{VH}$Y112 and $_{VH}$Y103) and LCDR1 ($_{VL}$Y36) (Fig 3A and 3D). Meanwhile, residues in the VP3 B-B knob ($_{VP3}$K61 and $_{VP3}$Y63) and HI-loop ($_{VP3}$Q197) interact with residues in light chain CDR3 (LCDR3) ($_{VL}$S97) and LCDR1 ($_{VL}$S28). The side chains of $_{VP2}$Q196 and $_{VP2}$D68 form hydrogen bond contacts with the $_{VH}$Y117 side chain. The side chains of $_{VP2}$T70 and $_{VP2}$T71 form hydrogen bond contacts with the $_{VL}$Y36 side chain. The side chain of $_{VP2}$K73, $_{VP2}$H77, $_{VP2}$E131 and $_{VP2}$K137 also forms hydrogen bond contacts with the side chain of $_{VH}$Y112, $_{VH}$R120, $_{VH}$Y56 and $_{VH}$Y103, respectively.

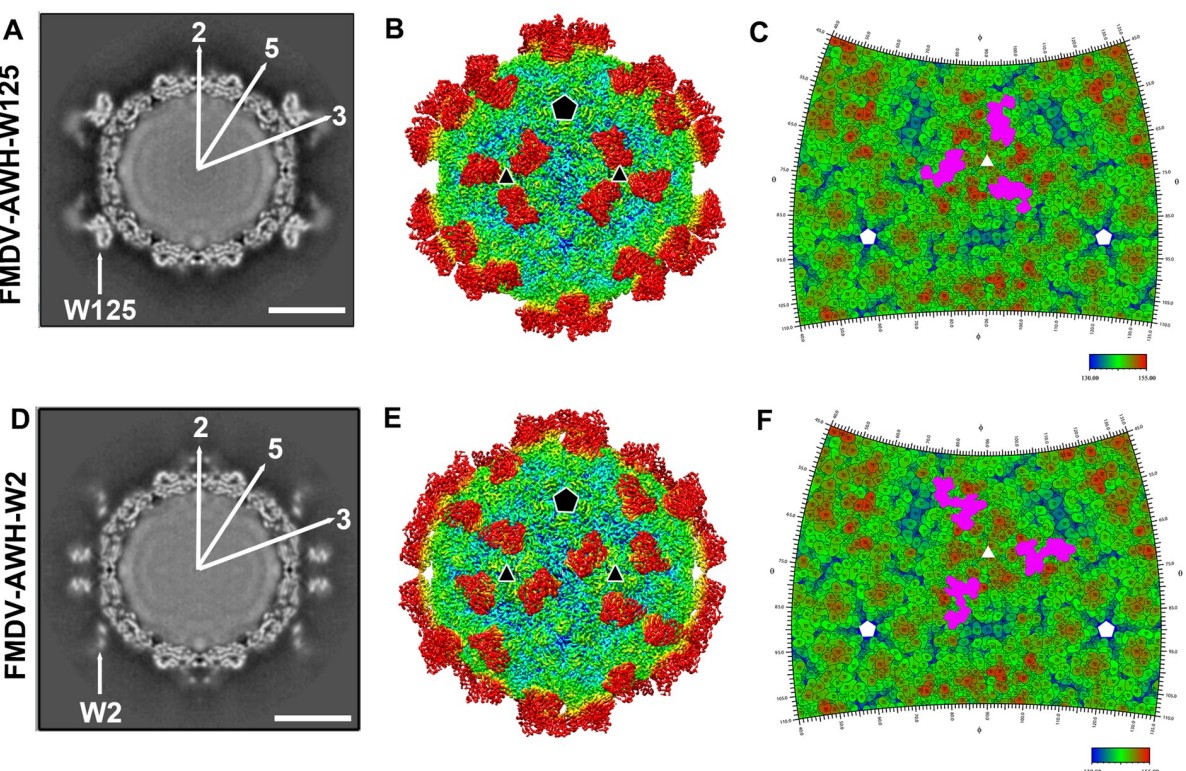

**Fig 2. Cryo-EM structures of the FMDV-AWH-W125 complex and FMDV-AWH-W2 complex.** The central cross-sections through cryo-EM maps of the FMDV-AWH-W125 complex (A) and FMDV-AWH-W2 complex (D) are shown with icosahedral 2-, 3- and 5-fold axes. Each image in the 480-pixel boxes corresponds to 446 Å in each dimension. Scale bars, 100 Å. Rendered images of the FMDV-AWH-W125 complex (B) and FMDV-AWH-W2 complex (E). Depth cueing with color is used to indicate the radius (< 120 Å, blue; 130–150 Å, from cyan to yellow; > 160 Å, red). The icosahedral five- and threefold axes are represented by pentagons and triangles, respectively. Footprints of W125 (C) and W2 (F) on the FMDV surface. Each figure shows a 2D projection of the FMDV surface produced using RIVEM [43]. The 5- and 3-fold icosahedral symmetry axes are marked as pentagons and triangles, respectively, on one icosahedral asymmetrical unit. The spherical polar angles (θ, φ) define the location on the icosahedral surface. The depictions are radially depth cued from blue (radius = 130 Å) to red (radius = 155 Å). The W125 and W2 footprints are pink-colored.

Meanwhile, the side chains of $_{VP3}$K61 and $_{VP3}$Y63 form hydrogen bond contacts with the $_{VL}$S97 side chain. The side chains of $_{VP3}$Q197 form hydrogen bond contact with the $_{VL}$S28 side chain (S1 Table). To further validate the crucial determinants of FMDV serotype A for W125, we substituted alanine for FMDV capsid residues involved in the virus-antibody complex interface. A total of 8 single-substitution mutants were successfully rescued (Fig 3E and S2 Table) and assessed for neutralization potency with W125. As shown in Fig 3F, mutations at positions 70, 71, 77 and 196 on VP2 as well as position 61 on VP3 significantly reduced antibody neutralization. Meanwhile, the sequence alignment of A/WH/CHA/09, A/GDMM/2013 and A/AF72 shows that the common residues ($_{VP2}$T70, $_{VP2}$H77, $_{VP2}$Q196, and $_{VP3}$K61) that contact W125 are strictly conserved (Fig 3G). The conservation of all residues appeared more than 95% in available FMDV serotype A sequences (S5 Fig), indicating that these highly conserved interaction residues may be the key determinants for conserved antigenic structure on FMDV serotype A.

Distinct with W125, W2 makes contact with the VP3 involving B-B knob, BC loop, βC, CD/EF/GH loops, revealing a conserved antigenic site on VP3 (Fig 4). Concretely, the residue ($_{VP3}$D59) in the VP3 B-B knob interacts with residue ($_{VL}$Y36) in LCDR1. Meanwhile, residues in VP3 BC-loop ($_{VP3}$R67, $_{VP3}$D69 and $_{VP3}$Q71), βC ($_{VP3}$K76), CD-loop ($_{VP3}$K84), EF-loop

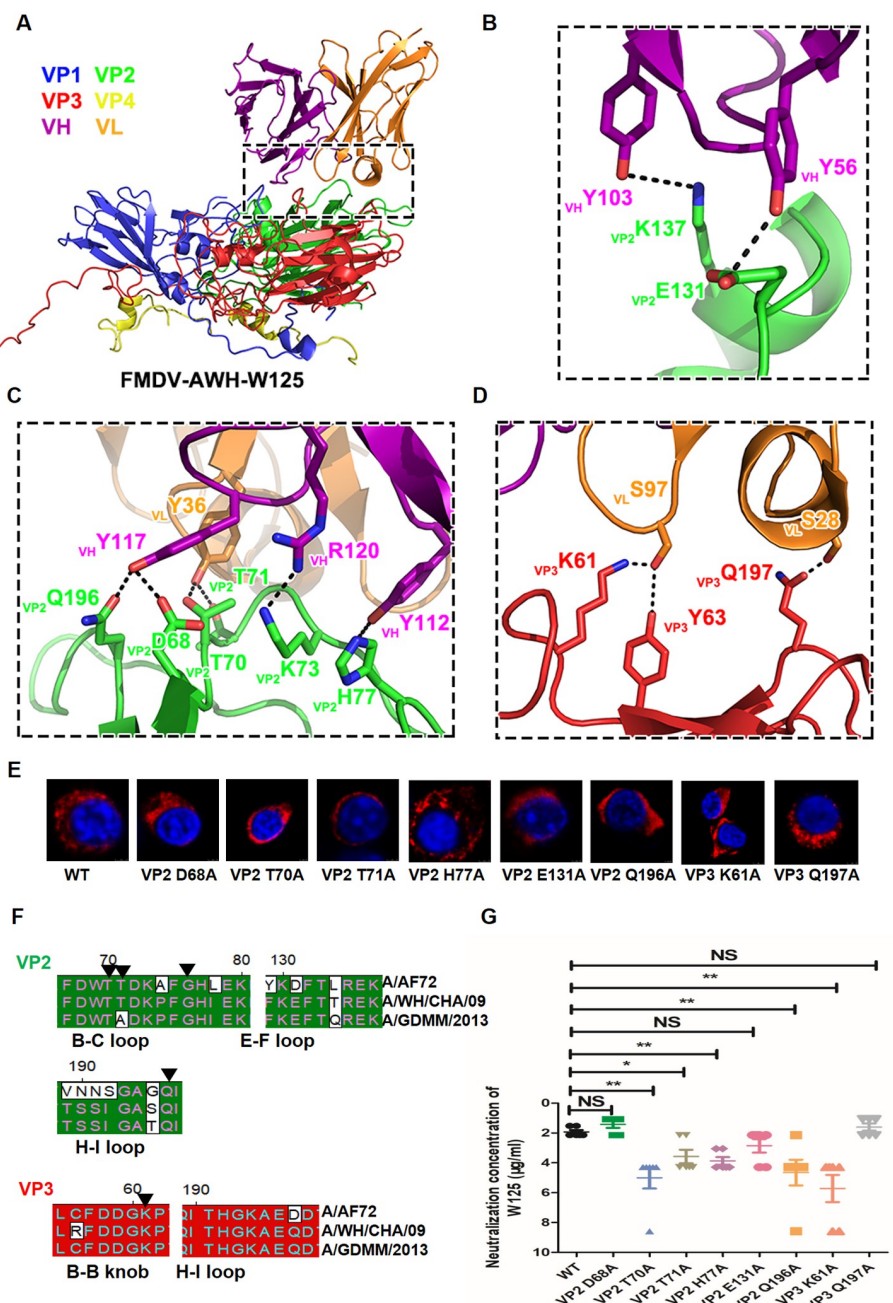

**Fig 3. Structure of the FMDV-AWH-W125 complex and key determinations on VP2 as well as VP3 of FMDV serotype A.** (A) Cartoon representation of one protomer showing the interaction interface between W125 scFv and the capsid. The heavy chain and light chain of W125 are colored purple and orange, respectively. The capsid proteins VP1 to VP4 are colored blue, green, red and yellow, respectively. (B to D) Expanded views of the interaction interface highlighting the Bβ, E-F loop (B), B-C loop and H-I loop (C) of VP2 as well as B-B knob and H-I loop (D) of VP3 within one protomer. Presumable hydrogen bonds and salt bridges in the interaction interface are marked by black dashed lines. (E) Identification of rescued single-substitution mutants by immunofluorescence analysis. BHK-21 cells were infected with rescue mutants at an MOI of 10 for 4 h. FMDV protein 3A was detected using mouse mAb 3A24 and an Alexa Fluor 561-conjugated secondary antibody. (F) Sequence alignment of VP2/VP3 of A/AF72, A/WH/CHA/09 and A/GDMM/2013 strains. The critical residues in interactive interfaces are indicated with black triangles. (G) The neutralization efficacy of W125 against wildtype (A/WH/CHA/09) and mutants corresponding to interactive residues (VP2 D68A, VP2 T70A, VP2 T71A, VP2 H77A, VP2 E131A, VP2 Q196A, VP3 K61A and VP3 Q197A) was evaluated using a microneutralization assay. The neutralization concentration represents the lowest antibody required to fully prevent CPE. * indicates a significant difference compared to wildtype at P<0.05. ** indicates a significant difference compared to wildtype at P<0.01. NS indicates no significant difference.

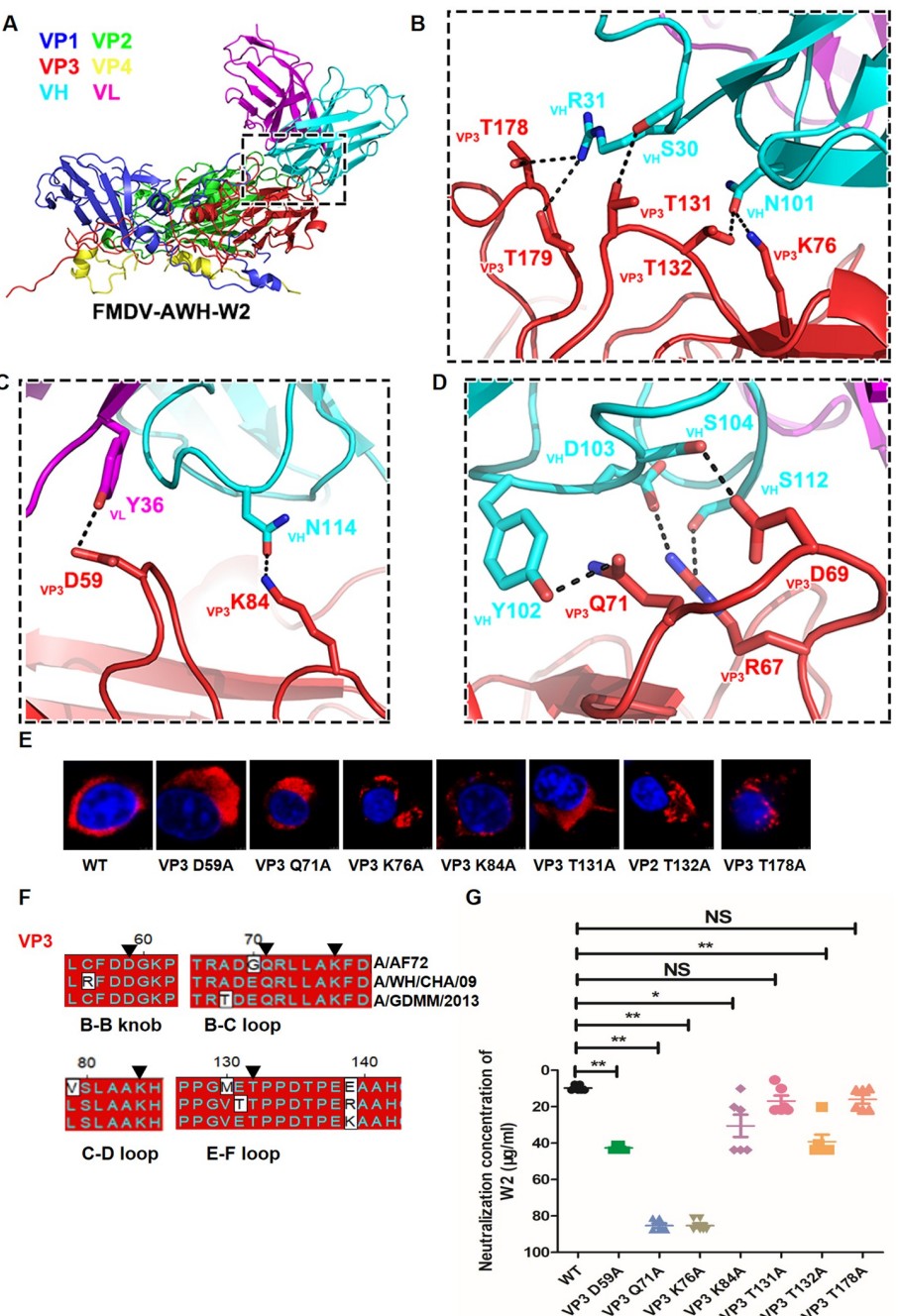

**Fig 4. Structure of the FMDV-AWH-W2 complex and key determinations on VP3 of FMDV serotype A.** (A) Cartoon representation of one protomer showing the interaction interface between W2 scFv and the capsid. The heavy chain and light chain of W2 are colored sky blue and rose madder, respectively. The capsid proteins VP1 to VP4 are colored blue, green, red and yellow, respectively. (B to D) Expanded views of the interaction interface highlighting the B-B knob, CD loop (C), βC, EF/GH loops (B), BC loop (D) on VP3. Presumable hydrogen bonds and salt bridges in the interaction interface are marked by black dashed lines. (E) Identification of rescued single-substitution mutants by immunofluorescence analysis. BHK-21 cells were infected with rescue mutants at an MOI of 10 for 4 h. FMDV protein 3A was detected using mouse mAb 3A24 and an Alexa Fluor 561-conjugated secondary antibody. (F) Amino acid sequence alignment of VP3 of A/AF72, A/WH/CHA/09 and A/GDMM/2013 strains. The critical residues in interactive interfaces are indicated with black triangles. (G) The neutralization efficacy of W2 against wildtype (A/WH/CHA/09) and mutants corresponding to interactive residues (VP3 D59A, VP3 Q71A, VP3 K76A, VP3 K84A, VP3 T131A, VP3 T132A and VP3 T178A) was evaluated using a microneutralization assay. The neutralization concentration represents the lowest antibody required to fully prevent CPE. * indicates a significant difference compared to wildtype at P<0.05. ** indicates a significant difference compared to wildtype at P<0.01. NS indicates no significant difference.

($_{VP3}$T131 and $_{VP3}$T132) and GH-loop ($_{VP3}$T178 and $_{VP3}$T179) interact with residues in HCDR3 ($_{VH}$D103, $_{VH}$S112, $_{VH}$S104, $_{VH}$N101, $_{VH}$N114 and $_{VH}$Y104) and HCDR1 ($_{VH}$S30 and $_{VH}$R31) (Fig 4A and 4D). The side chains of $_{VP3}$K76 and $_{VP3}$T132 form hydrogen bond contacts with the $_{VH}$N101 side chain. The side chains of $_{VP3}$T178 and $_{VP3}$T179 form hydrogen bond contacts with the $_{VH}$R31 side chain. The side chain of $_{VP3}$D59, $_{VP3}$R67, $_{VP3}$D69, $_{VP3}$Q71, $_{VP3}$K84 and $_{VP3}$T131 also forms hydrogen bond contacts with the side chain of $_{VL}$Y36, $_{VH}$D103, $_{VH}$S104, $_{VH}$Y102, $_{VH}$N114 and $_{VH}$S30, respectively (S3 Table). For illustration of the key determinants on VP3, we further substituted alanine for FMDV capsid residues involved in the FMDV-AWH-W2 complex interface. A total of 7 single-substitution mutants were successfully rescued (Fig 4E and S2 Table) and assessed for neutralization potency with W2. As shown in Fig 4F, mutations at positions 59, 71, 76, 84 and 132 on VP3 obviously reduced antibody neutralization; in particular, VP3 residue 71 and 76 mutations resulted in a significant reduction (~10-fold) in the virus-neutralizing (VN) titer. Meanwhile, the sequence alignment of A/WH/CHA/09, A/GDMM/2013 and A/AF72 shows that the common residues ($_{VP3}$D59, $_{VP3}$Q71, $_{VP3}$K76, $_{VP3}$K84 and $_{VP3}$T132) that contact W2 are strictly conserved (Fig 4G). Further analysis of available FMDV serotype A sequences revealed the $_{VP3}$K76 and $_{VP3}$K84 were extremely constant with conservation of 99% and 100% (S5 Fig), indicating that the two highly conserved interaction residues represent key determinants for conserved antigenic structure on VP3 of FMDV serotype A.

FMDV employs integrin (generally avβ6) as primary receptor to entry epithelial cells, causing infection in susceptible host [23,24]. FMDV binding to the integrin receptor is facilitated by a conserved arginine-glycine-aspartic (RGD) motif in the exposed GH-loop of VP1 [24]. Structure comparisons of FMDV-integrin and FMDV-mAbs show obvious clashes between mAbs (W2 and W125) and the integrin receptor, suggesting that FMDV neutralization by W2 and W125 is facilitated by blocking virus-receptor interaction via steric hindrance (S6 Fig).

## The conserved and key determinants on VP1 of FMDV serotype A

To explore other conserved antigen sites out of VP2 and VP3 on FMDV serotype A, the neutralization escape mutants were selected for the remaining bnAbs (W145, W153 and W151) derived from bovine. As shown in Table 3, these neutralization escape mutants reveled key determinants on VP1, involving residues on the VP1 C-D loop, G-H loop and C-terminus. Concretely, the position 58 on C-D loop and the position 147 on G-H loop were respectively key antigenic determinants representing antigen sites 3 and 5 on FMDV serotype A, as shown by the W153-

**Table 3. Bovine broad neutralizing mAb escape mutants.**

| MAb | Parent virus | Frequency of mutants[$] | Residue change | Neutralization Concentration[#] (µg/mg) | Antigenic site |
|---|---|---|---|---|---|
| **W145** | A/WH/CAH/09 | 3/6 | VP1 G147E | 600 | Site 5 |
| | | 1/6 | VP1 G147V | 600 | |
| | | 1/6 | VP1 S148P, L149P | 600 | |
| | | 1/6 | VP1 G147E, S148P | 600 | |
| **W151** | A/WH/CAH/09 | 2/4 | VP1 G147E; VP2 D72E | 600 | Site 5 and site 2 |
| | | 2/4 | VP1 G147E; VP2 K73R | 600 | |
| **W153** | A/WH/CAH/09 | 2/3 | VP1 Q58R, K209Q | 600 | Site 3 |
| | | 1/3 | VP1 Q58R | 600 | |

[#]Neutralization concentration was determined as the lowest antibody concentration that protected cells from CPE.

[$]Frequencies of the mutants are the number of mutants with the mutation at the indicated residue/total number of mutants obtained.

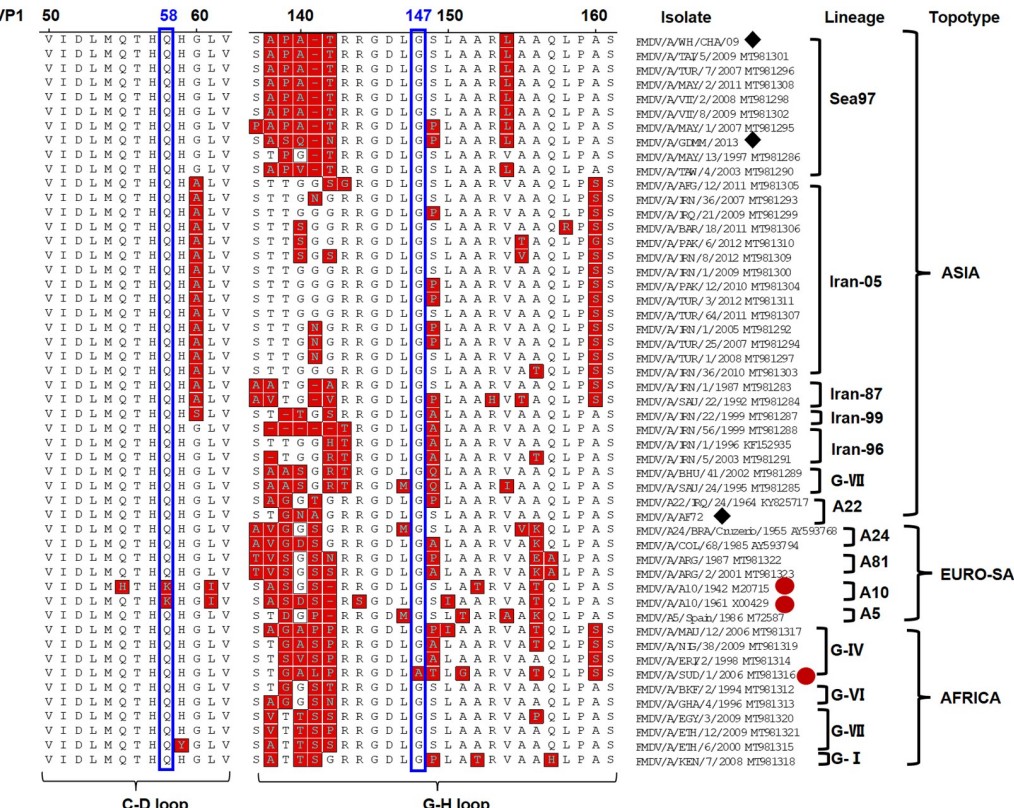

**Fig 5. Sequence alignment of VP1 of representative FMDV serotype A strains in ASIA, EURO-SA and AFRICA topotypes.** The VP1 amino acids sequences of A/AF72, A/WH/CHA/09 and A/GDMM/2013 strains (marked with diamond) were aligned with that of 48 serotype A strains retrieved from NCBI, including sevens lineages from ASIA topotype, four lineages from EURO-SA topotype and four lineages from AFRICA topotype. The identified key determinations at positions 58 and 147 were circled with blue box and the strains with varied residues involving determinations were indicated with red circle.

and W145-escape mutants. In addition, two W153-escape mutants (2/3) had substitutions on both 58 and 209 residues on the VP1, indicating the interaction between the VP1 C-D loop and C-terminus for forming antigen site 3. Interestingly, all the W151-escape mutants displayed two distant substitutions that separately located in the VP2 72/73 residues on B-C loop corresponding to antigen site 2 as well as the VP1 147 residue on G-H loop corresponding to antigen site 5, this indicated bovine bnAb W151 could recognize a novel epitope across two different antigen sites, suggesting flexible G-H loop on VP1 approaching to B-C loop of VP2 on FMDV serotype A. Moreover, by alignment of the viral capsid sequence of A/AF72, A/WH/CHA/09 and A/GDMM/2013 strains with other representative strains from ASIA, EURO-SA, and AFRICA topotypes, it was found that the substitutions identified in neutralization-escape mutants were highly conserved residues on these strains, indicating key determinants including the VP1 58/147 represent the conserved antigen sites on VP1 of FMDV serotype A (Fig 5 and S5 Fig).

## The shrinkage and appearance of strain-specific antigen epitopes were found on VP3 68 and 175 positions of FMDV serotype A

To reveal the antigenic variation of FMDV serotype A under immune pressure, we further performed the screening of neutralization-escape mutants using 20 strain-specific antibodies

against A/AF72 as well as 19 strain-specific antibodies against A/GDMM/2013. For the historical A/AF72 strain, the results showed most (11/20) of the A/AF72 and/or A/WH/CHA/09-NAbs driven the variation at position 68 on VP3 of A/AF72 strain with AA mutant from A to T/V (Fig 6A and S4 Table). Alignment of the VP3 of A/AF72, A/WH/CHA/09 and A/GDMM/2013 strains indicated the position 68 (A) residue was constant on the A/AF72 and A/WH/CHA/09 strains, but the appeared variation (A→T) on the A/GDMM/2013 strain was consistent with the mutants pressured by A/AF72-NAbs. These results indicated the key 68 determinant represent a strain-specific epitope for A/AF72 or A/WH/CHA/09. Contrastingly, only one NAb (R136) available in the A/GDMM/2013-NAbs (1/19) was identified to recognize this site. Thus, we concluded that the A/GDMM/2013 strain might produce naturally occurring shrinkage of the epitope under evolution to adapt the changing environment. For reconstruction of the epitope on A/GDMM/2013 strain, we successfully rescued the VP3 68 (T→A)

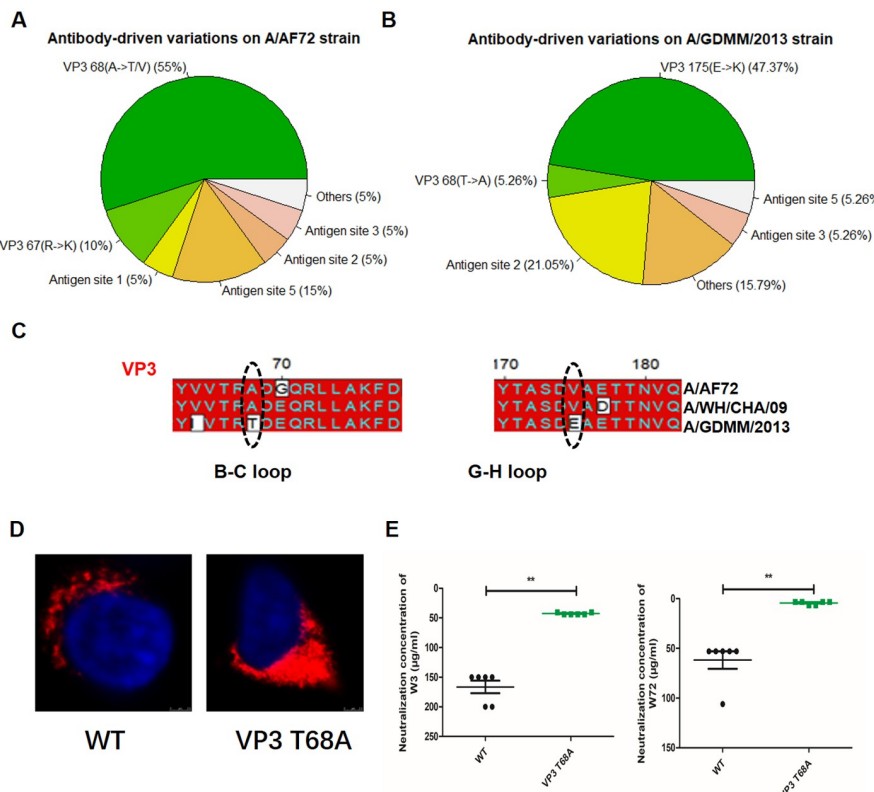

**Fig 6. The strain-specific epitopes identified on FMDV serotype A A/AF72 strain and A/GDMM/2013 strain.** The antibody-driven variations on A/AF72 strain (A) and A/GDMM/2013 strain (B) were separately determined by selection of neutralization-escape mutants using 20 A/AF72-specific and A/GDMM/2013-specific neutralizing mAbs individually. The proportion of pin chart indicated each mAb-driven variation accounted for the five known antigen sites encompassing VP1 GH loop and C-terminus (site 1/5), VP1 B-C loop (site 3), VP2 B-C loop (site 2) and VP3 B-B knob (site 4), as well as the other unidentified site. (C) Amino acid sequence alignment of VP3 of A/AF72, A/WH/CHA/09 and A/GDMM/2013 strains. The VP3 68 and 175 positions that formed strain-specific epitopes were framed with black oval circles. (D) Immunofluorescence analysis of rescued VP3 68 (T→A) mutant that was constructed basis on entire P1 gene of A/GDMM/2013 strain. BHK-21 cells were infected with the rescue mutant or wildtype virus (A/GDMM/2013) at an MOI of 10 for 4 h. FMDV protein 3A was detected using mouse mAb 3A24 and an Alexa Fluor 561-conjugated secondary antibody. (E) The neutralization efficacy of the A/AF72-specific mAbs W3 and W72 against wildtype (A/GDMM/2013) and its mutant (VP3 T68A) was evaluated using a microneutralization assay. The neutralization concentration represents the lowest antibody required to fully prevent CPE. ** indicates a significant difference compared to wildtype at P<0.01.

mutant based on whole structural proteins of A/GDMM/2013 strain using reverse genetic technique (Fig 6D). The obtained A/GDMM/2013 VP3 68 (T→A) mutant was expectedly neutralized by the A/AF72-specific NAbs W3 and W72 (Fig 6E). This indicated the flexibility and operability of the epitope, revealing a novel target for expansion of serotype A antigen spectrum.

For the latest A/GDMM/2013 strain, most (9/19) of the A/GDMM/2013-NAbs drive the variation at position 175 on VP3 of A/GDMM/2013 strain, exhibiting AA mutant from the negative charged E to the positive charged K, and this variation may disrupt the charge interaction and enable escape antibody neutralization (Fig 6B and S4 Table). Alignment of the VP3 AA sequences of the three strains showed the position 175 (V) was common on A/AF72 and A/WH/CHA/09 strains, appearing variation (V→E) on the A/GDMM/2013 strain. Contrasting with the position 175 variation (E→K) on most of A/GDMM/2013-resistant mutants, we did not observe variation on this position in all the obtained A/AF72-resistant mutants. Thus, we speculated the key 175 determinant was a newly appearing strain-specific epitope for the A/GDMM/2013, representing a feature of antigenic evolution of FMDV serotype A with time migration.

## Antibody-driven variations on FMDV serotype A exhibited the distribution difference on viral capsid proteins between A/AF72 and A/GDMM/2013

To further reveal molecular basis of antigenic variation between A/AF72 and A/GDMM/2013, we made a comparative analysis of the frequency and position of mutant residues on viral capsid proteins of the strain-specific neutralization-resistant mutants (S4 Table). As shown in Fig 7, the mutation of the A/AF72 strain was mainly distributed on VP3 (Fig 7A), involving 15 residues in VP3 B-B knob ($_{VP3}$59), B-C loop ($_{VP3}$65, $_{VP3}$67, $_{VP3}$68 and $_{VP3}$71), βE ($_{VP3}$119), E-F loop ($_{VP3}$130), G-H loop ($_{VP3}$177), H-I loop ($_{VP3}$196 and $_{VP3}$197), and the C-terminus ($_{VP3}$202, $_{VP3}$203, $_{VP3}$205, $_{VP3}$206 and $_{VP3}$207); secondary distributed on VP1 (Fig 7B), involving 9 residues in VP1 C-D loop ($_{VP1}$58), E-F loop ($_{VP1}$99) G-H loop ($_{VP1}$147- $_{VP1}$151 and $_{VP1}$153) and C-terminus ($_{VP1}$198); and the least distributed on VP2 (Fig 7C), involving only 5 residues in VP2 βB ($_{VP2}$65), B-C loop ($_{VP2}$71 and $_{VP2}$72), H-I loop ($_{VP2}$190) and βI ($_{VP2}$198). The proportion of mutant residues on different structural proteins of A/AF72 suggested that VP2 was relatively constant while VP3 was more tolerable variation under immune pressure. Distinct from the A/AF72, A/GDMM/2013 occurred more variations on VP1, as revealed by half of total mutant residues distribution in VP1 N-terminus ($_{VP1}$13), B-C loop ($_{VP1}$43, $_{VP1}$46 and $_{VP1}$48), C-D loop ($_{VP1}$58), βD ($_{VP1}$81), D-E loop ($_{VP1}$84), E-F loop ($_{VP1}$99), G$_1$-G$_2$ loop ($_{VP1}$122 and $_{VP1}$123), G-H loop ($_{VP1}$147- $_{VP1}$149 and $_{VP1}$158), H-I loop ($_{VP1}$170 and $_{VP1}$172) and the C-terminus ($_{VP1}$193) (Fig 7D). The numbers of mutant residues in VP2 and VP3 were similar, displaying 7 residues on VP2 B-C loop ($_{VP2}$72 and $_{VP2}$74), βC ($_{VP2}$79), E-F loop ($_{VP2}$130, $_{VP2}$134 and $_{VP2}$137) and G-H loop ($_{VP2}$171) as well as 8 residues on VP3 N-terminus ($_{VP3}$25), B-B knob ($_{VP3}$56 and $_{VP3}$61), B-C loop ($_{VP3}$68), G-H loop ($_{VP3}$174- $_{VP3}$175) and C-terminus ($_{VP3}$195 and $_{VP3}$220) (Fig 7E and 7F). Overall, these variations above reflected the antigenic diversity of different strains and described an antigenic characterization of FMDV serotype A, exhibiting a relatively stable VP2 and the tolerable variability for VP3 and VP1.

Notably, projecting these mutant residues on surface of FMDV serotype A revealed that strain-specific epitopes were mapped within or near the conserved antigen sites on VP2 and VP1 of FMDV serotype A, displaying part overlap (S4 Table). Such as, the A/AF72 or A/AF72-specific mAbs (W121, R55, R95 and R127) driven the variations at positions 71, 72, 73, 74, 79, 82, 130, 131 and 196 on VP2 involving BC/EF/HI loops (antigen site 2), which was also

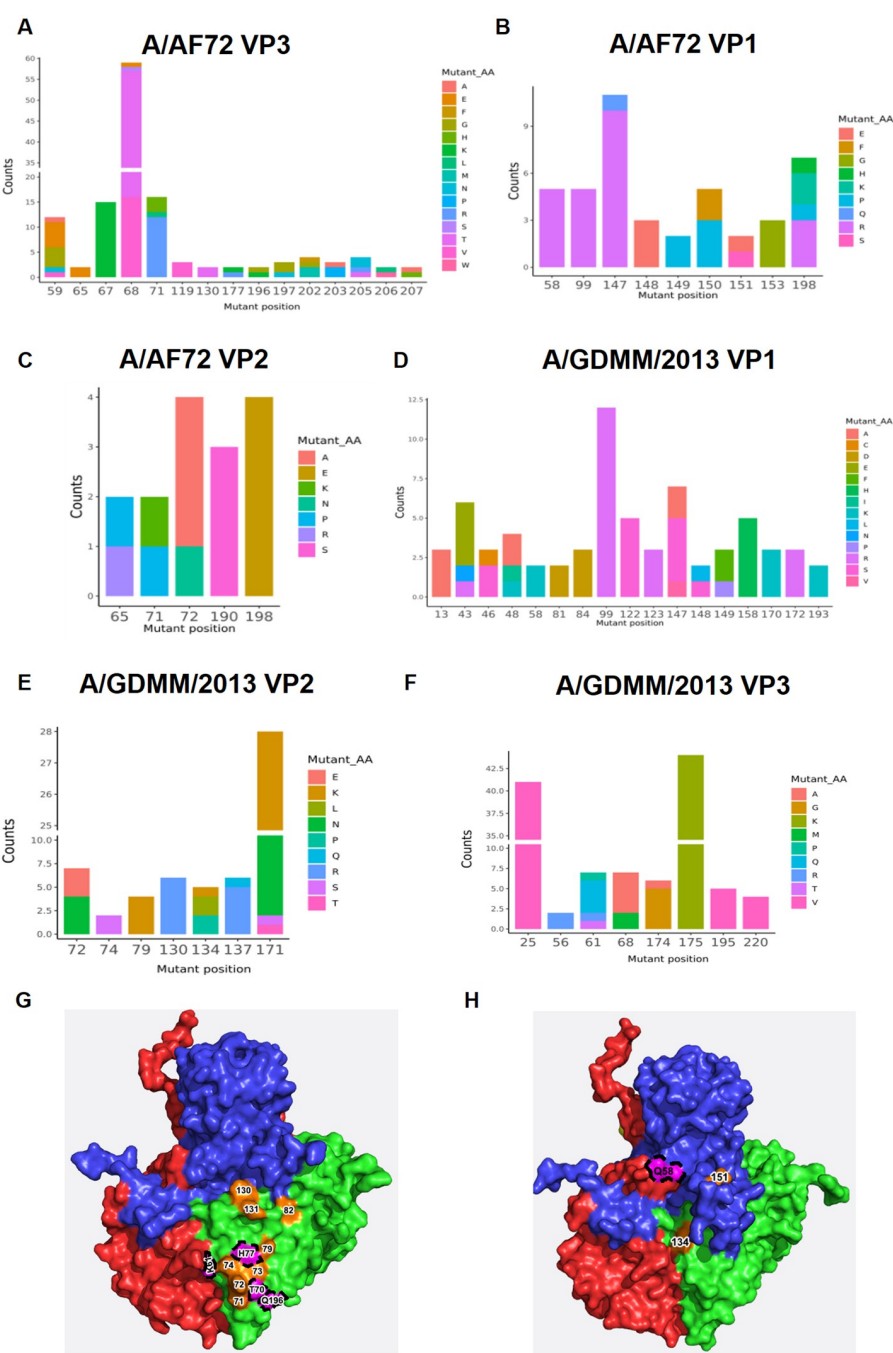

**Fig 7. The distribution characterization of antibody-driven variations on viral capsid surface of FMDV serotype A between A/AF72 strain and A/GDMM/2013 strain.** A total of 135 A/AF72-neutralizaiton escape mutants were obtained by antibody pressure selection using 20 A/AF72-specific neutralization antibodies. The antibody-driven variations on A/AF72 strain were identified by sequence alignment of the 135 mutants and the frequency of varied amino acid residues at positions on VP3 (A), VP1 (B) and VP2 (C) were separately displayed. A total of 117 A/GDMM/2013-neutralizaiton escape mutants were obtained by antibody pressure selection using 19 A/GDMM/2013-specific neutralization antibodies. The antibody-driven variations on A/GDMM/2013 strain were identified by sequence alignment of the 117 mutants and the frequency of varied amino acid residues at positions on VP1 (D), VP2 (E) and VP3 (F) were separately displayed. (G) Footprint of key determinations on VP2 recognized by broad neutralization antibody (W125) and strain-specific mAbs (W121, R55, R95 and R127). The conserved determinations (VP2T70, VP2H77, VP2Q196 and VP3K61) recognized by W125 were circled with black dotted line and highlighted with pink. The surfaced determinations recognized by W121 (VP271 and VP272), R55 (VP271, VP279 and VP282), R95 (VP271, VP272, VP282 and VP2131), and R127 (VP272, VP273, VP274, VP2130 and VP2196) were all indicated in orange. (H)

Footprint of key determinations on VP1 recognized by broad neutralization antibody W153 ($_{VP1}$58) and strain-specific mAb W155 ($_{VP1}$58, $_{VP1}$151 and $_{VP2}$143). The capsid proteins VP1, VP2 and VP3 on one protomer are colored blue, green and red, respectively.

targeted by bovine bnAb W125 (Fig 7G). On VP1, the key determination 58 residue was shared in both conserved and varied epitopes that were recognized by bovine bnAb W153 and A/AF72-specific NAb W155 (Fig 7H). These results suggested the antigenic sites on FMDV serotype A were consisted of conserved and strain-specific epitopes that were spatially unseparated. However, the distribution difference of antibody-driven variations suggested the immunodominance of each epitope might differ and thus showed the different antigenicity among lineages of FMDV serotype A.

## Discussion

This study describes the first development of bovine-derived monoclonal NAbs against different period strains of FMDV serotype A and their conserved antigen structures on VP2 and VP3 of virus capsid revealed by Cryo-EM. Bovine NAbs revealed at least four conserved antigen sites including two sites on VP1 and each one on VP2/VP3, which exist on viral capsid surface and can induce bnAb response to FMDV serotype A in vivo. Additionally, the key determinants at positions 68 and 175 on VP3 consisting of strain-specific epitopes were spatially separate and mapped on two flanks of the conserved antigen epitopes, depicting the antigenic variations on FMDV serotype A (Fig 8).

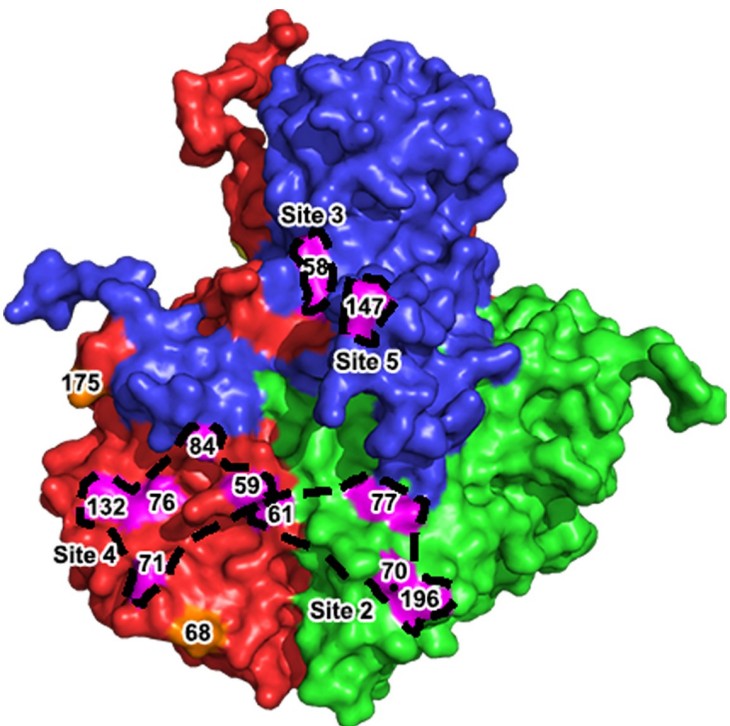

**Fig 8. The conservative and variable antigen sites of FMDV serotype A recognized by bovine neutralization antibodies.** Four conservative antigen sites were indicated with pink and respectively determined by bnAb W125 (site 2), W153 (site 3), W2 (site 4) and W145 (site 5). The variable antigen sites involving the position 68 and 175 on VP3 that were recognized by A/AF72-specific and A/GDMM/2013-specific antibodies were marked with oranges.

In this work, there are two limitations to the immunization schedule for accurately evaluating the antibody response to FMDV serotype A in bovine. First, regarding the potential for "original antigenic sin", previous sequential immunizations with O serotype antigens in bovine may interfere with the quality and magnitude of neutralizing antibodies responses against serotype A. Second, this study lacks the sera data collected from different times after sequential immunizations with A antigens, so we cannot rule out that only one of the three immunizations were effective and that the sera simply cross-reacted to neutralize all three serotype A strains.

FMDV VP1 is highly accessible to the host immune system and contains a receptor binding site (RGD motif) in the G-H loop, which can undergo continuous variation under immune pressure, leading to the emergence of new epidemic strain. Thus, the VP1 G-H loop was recognized to be most flexible on surface of FMDV. For serotype A, the amino acid depletion before RGD motif was observed in some lineage strains and caused the variance in length of VP1. The residue VP1 149 (P), located at the position +3 after the RGD motif, was strongly associated with a match to vaccine A22/IRQ/24/64 and speculated to be a key antigen determination representing on subset of serotype A strains [25]. Our study showed that the VP1 147 (G), corresponding to the residue at the position +2 after the RGD motif, was a conserved antigen determination that could induce broad neutralization antibody (W145) against both the A22 and Sea97 lineage strains. In addition, the residue at position +2 after the RGD motif was also identified as key antigen determination on A24 Cruzeiro strain [14]. Given to the conservation of residues after the RGD motif, we concluded that the G-H loop might contain multiple epitopes that residue (G) at position +2 after the RGD motif was a conserved antigen determination for serotype A, whereas the residue at position +3 after the RGD motif could represent a strain or lineage specific antigen determination. The antigenic structure resolved by FMDV-AWH-W125 complex indicate the conserved antigenic site 2. The antigenic site 2 containing epitopes have also been identified on other topotypes of serotype A isolates, including the Europe isolates A5 [10] and A10 [11], as well as isolates from the East Africa [26], and this suggests the structural conservation of this site on FMDV serotype A. The interface residues in antigenic site 2 solely focused on B-C/E-F/H-I loops of VP2 for serotype O, as resolved in virus-antibody complex structures in our previous report [8,9]. However, the FMDV-AWH-W125 complex structure revealed an additional key 61 residue on the VP3 B-B knob, along with the interface residues on B-C/H-I loops of VP2. This represents a structural difference in antigenic site 2 between FMDV serotype A and O.

The antigenic epitopes on VP3 were identified in different FMDV serotype A isolates, containing the A12 isolate and the A10 isolate from Europe [11,27,28]. The distribution proportion of mAbs targeting to viral capsid proteins of serotype A in the documents suggested that VP3, which encompasses an antigenic site, may be the immunodominant. Our study showed appropriately 52% (23/44) of total bovine neutralizing mAbs recognized the epitopes on VP3 and this indicated the immunodominance of VP3 on serotype A. However, the immunodominant antigenic site on serotype A could be variable and depend on different strains. For A/AF72 strain, the VP3 showed obvious immunodominance and was targeted by almost 70% (14/20) of the strain-specific mAbs. Regarding the A/GDMM/2013 strain, the VP1 and VP3 encompassing epitopes were interconnected and immunodominant due to their recognition by half (10/19) of the strain-specific mAbs. Although the conserved antigenic structure was also identified on VP3 of serotype A, the contrastive analysis of the interface in W2-A/WH/CHA/09 and C4-O/Tibet/99 complex structures showed a significant difference in antigenic structure on VP3 of FMDV between serotype A and O [9]. Structurally, there exists an inter-protomer antigen structure on serotype O, covering the B-B 'knob', βB, B-C loop, E-F loop and H-I loop of VP3 and B-C loop of VP2, as well as the H-I loop of VP2 from adjacent

protomer. As for serotype A, it is a concentrated antigen structure involving B-B 'knob', B-C, E-F and G-H loops on VP3 of serotype A. In addition, the key determinations of the antigen site between the two serotypes were inconsistent, displaying the T65, T68, E131, K134 and G196 on VP3 of serotype O versus the D59, Q71, K76, K84 and T132 on VP3 of serotype A.

The evolution of virus is influenced by its existing ecological factors, such as the viral fitness to different hosts and pre-existing immunity [29]. FMDV is subject to continuous evolution under immune pressure, giving rise to extensive genetic and antigenic variation within each serotype. In this study, bovine NAbs drive the A/AF72 strain (A22 lineage) VP3 68 position mutation from A to T, meanwhile this mutant residue appeared in A/GDMM/2013 strain in Sea97 lineage. This evidence the pre-existing immunity in bovine could be a main driver of antigenic evolution of FMDV serotype A from the A22 lineage to SEA97 lineage. The VP3 175 (E) position represents a newly emerging epitope on A/GDMM/2013 strain and also appears in the sub-lineage of A-IR05 [30]. These diverse epitopes on VP3 could affect the width of the antigenic spectrum of FMDV serotype A. It is important to note that it was the A/AF72 strain, not the A/GDMM/2013 strain, that elicited bnAbs secretory B cells against three representative strains in both A22 and Sea97 lineages. This suggests that the A/AF72 remains a good vaccine candidate for defending against currently epidemic isolates from the Sea97 lineage.

Vaccination is universally recognized as the primary strategy for eradicating of FMDV. Under immune pressure, FMDV evolution drive the distribution unbalance of conserved and strain-specific antigen sites on viral capsids surface, and biased to strain-specific antigen sites led to immune escape. Therefore, broad antigen spectrum coverage was a necessary trait for good FMD vaccine development. Through full vaccination, FMD serotype A were well controlled, whereas serotype O were still sporadically occurred in China. This could be explained by distribution difference of conserved and strain-specific antigen sites between FMDV serotypes. Comparing with previous data, more conserved antigen sites (four) existed on serotype A than that (only two) on serotype O. In addition, the most strain-specific antigen sites were identified on VP1 for serotype O versus that on VP3 for serotype A.

In summary, using bovine broad and strain-specific neutralization antibodies, we revealed four conserved antigen sites existing on FMDV serotype A and compared structure difference on VP2 and VP3 between serotype A and O. Strain-specific antibodies illustrated the antigenic evolution on VP3 of FMDV serotype A under immune pressure. This study provided antigenic information to select candidate vaccine strain and guide the design of broad vaccine molecular against FMDV serotype A.

## Material and methods

### Ethics statement

All the animal experiments in the present study were approved by the Review Board of Lanzhou Veterinary Research Institute, Chinese Academy of Agricultural Sciences (Permit No. LVRIAEC2018-006) and conducted in accordance with the Animal Ethics Procedures and Guidelines of the People's Republic of China on animal use.

### Bovine PBMCs

One-year-old healthy Qinchuan bovine (*Bos Taurus*), a Chinese breed of beef bovine, were raised in a clean animal room for the isolation of PBMCs after sequential vaccination with inactivated FMDV vaccine in this study. Briefly, bovine had been previously sequentially immunized with the separate three topotypes of FMDV serotype O (inactivated O/Mya/98, O/HN/CHA/93, and O/Tibet/99) as described in our previous report [21]. Then the bovine, designed as #1217, was further sequentially immunized with the inactivated vaccines of

FMDV A/WH/CHA/09, A/GDMM/2013 and A/AF72 at 28-day intervals. All vaccines were formulated with 146S antigen and Montanide ISA 201 (Seppic, Shanghai, China) by homogenization to obtain 6 µg antigen per dose in 2 mL vaccine. Animal was intramuscularly inoculated in the left side of the neck. After the last vaccination, the EDTA anticoagulation blood was sampled from peripheral of bovine and laid on HISTOPAQUE 1.083 (Sigma-Aldrich, USA) for isolation of PBMCs by centrifugation at 1200×g.

## Sorting of FMDV serotype A-specific B cells from PBMCs

FMDV A/GDMM/2013 or FMDV A/AF72 was propagated in baby hamster kidney BHK-21 cells. Following inactivation with BEI (Binary ethylenimine, BEI), approximately 1 liter of the supernatant containing virus antigens were precipitated by incubating at 4°C overnight in a solution of 8% (w/v) PEG 6,000. The resulting precipitated virus antigens were harvested by centrifugation at 3,500 g for 1 h at 4°C, and subsequently resuspended in 50 ml PBS (137 mM NaCl, 2.7 mM KCl, 50 mM $Na_2HPO_4$ and 10 mM $KH_2PO_4$, pH = 7.6). Next, viral antigens were pelleted through a cushion of 30% (w/v) sucrose in PBS by centrifugation at 35,000×g for a duration of 4 h. After removing the sucrose from the pellet, it was covered with 500 µl of PBS was added to cover the pellet. The supernatant was further purified over a 20–60% sucrose gradient and fractionated by centrifugation at 35,000 g for 4 h at 4°C. The fractions were then subjected to negative-stain electron microscopy analysis, and the fraction containing 146S particles was transferred to a 100-kDa MWCO centrifugal filter for buffer exchange with PBS to remove the sucrose.

Both of highly purified A/AF72 146S antigen and A/GDMM/2013 146S antigen were biotinylated with EZ-Link NHS-LC-biotin reagent (Thermo Fisher Scientific, USA) following the manufacturer's instructions. The resulting biotinylated FMDV 146S antigens (biotin-A/AF72 and biotin-A/GDMM/2013) were respectively used as the bait antigen for binding antigen-specific B cells. Fresh PBMCs were initially stained with biotin-A/AF72 or biotin-A/GDMM/2013, in combination with anti-bovine CD21-PE (Bio-Rad, USA) and anti-bovine IgM-FITC (Bio-Rad, USA, labeled with FITC in-house) for 30 min at 4°C in PBS buffer containing 2 mM EDTA and 0.5% BSA. Subsequently, a second-step antibody, mouse anti-biotin APC (Miltenyi Biotec, Germany), was added and incubated for 20 min at 4°C. Parallel staining of PBMCs lacking biotinylated FMDV 146S served as fluorescence minus one (FMO) control. These stained samples were immediately loaded on flow cytometry with a 100 µm nozzle (BD FACS Aria II, USA) and one million PBMCs were acquired to determine the proportion of FMDV-specific B cells. The $A/AF72^+IgM^-CD21^{+/-}$ and $A/GDMM/2013^+IgM^-CD21^{+/-}$ events were respectively sorted into 96-well plate at one cell per well. Subsequent single-cell PCR amplification and cloning of variable region genes (VH and VL) of bovine IgG followed our previously described protocol [21].

## Preparation of bovine mAbs

The paired VH and VL genes were synthesized with codon optimization for *Cricetulus griseus* and cloned into bovine heavy chain (CH-pcDNA3.4) and light chain (CL-pcDNA3.4) expression vectors, resulting in the antibody-expressing plasmids VH-CH-pcDNA3.4 and VL-CL-pcDNA3.4, respectively, for production of bovine IgG format antibody. The recombinant single-chain fragment variable (scFv) was designed by joining VH and VL fragments with a flexible linker (GGGGSGGGGSGGGGS) and adding a C-terminal His tag (HHHHHH). The optimized scFv gene was then cloned into the pcDNA3.4 expression vector. The scFv or the antibody expressing plasmids with a light-to-heavy chain ratio of 3:2 was transfected into suspended CHO cells (Invitrogen, USA) and cultivated for 10 days. The expressed mAbs in

supernatants were initially purified using Ni-chelating affinity chromatography and further purified by size exclusion chromatography using a Superdex 200 increase 10/300 column in an AKTA plus protein purification system (GE Life Sciences). The concentration of expressed mAbs was determined by measuring the absorbance values at a wavelength of 280 nm (A280).

### Virus neutralization test

The bovine mAbs were titrated for their viral neutralizing activity against A/AF72 stain (A22 lineage), A/WH/CHA/09 and A/GDMM/2013 strains (Sea97 lineage) from the ASIA topotype of FMDV serotype A as well as the rescued virus by using a micro-neutralization assay as previously described [21]. Briefly, antibody samples were serially diluted in 96-well cell culture plates at a 2-fold dilution rate, with a total volume of 50 μl. Then, 100 $TCID_{50}$ of FMDV in 50 μl of culture media was added to each well. After incubation for 1 h at 37˚C, approximately $5\times10^4$ BHK-21 cells in 100 μl media were added to each well as indicators of residual infectivity. Normal cell wells and virus control wells (0.1, 1, 10 and 100 $TCID_{50}$) in duplicate were included in each plate for comparison purpose. The plates were then incubated at 37˚C under 5% $CO_2$ conditions for 48 h before observing cytopathic effect (CPE). The experimental results were acceptable when complete CPE and no CPE appeared separately in 0.1 $TCID_{50}$ and 100 $TCID_{50}$ virus control wells. The endpoint titers were calculated by determining the reciprocal value of the last serum dilution to neutralize 100 $TCID_{50}$ FMDV in 50% of the wells. Neutralizing activity is expressed as the VN titer which is calculated as the initial antibody concentration divided by the endpoint titer.

### Cryo-EM sample preparation and data collection

FMDV 146 S and scFv were incubated at a molar ratio of 1:240 in a volume of 50 μl for 30 s at 4 ˚C. A 3-μl aliquot of the mixture was applied to a glow-discharged carbon-coated gold grid (GIG, Au 1.2/1.3, 200 mesh; Lantuo). The grid was blotted for 5 s in 100% relative humidity and plunge-frozen in liquid ethane using a Vitrobot mark IV (Thermo Fisher, USA). Cryo-EM data were collected at 200 kV with an FEI Arctica (Thermo Fisher, USA) and a direct electron detector (Falcon II, Thermo Fisher) at Tsinghua University. Micrograph images were collected as movies (19 frames, 1.2 s) and recorded at −2.4 to −1.4 μm under focus at a calibrated magnification of ×110 kX, resulting in a pixel size of 0.93 Å.

### Image processing and three-dimensional reconstruction

Individual frames from each micrograph movie were aligned and averaged using MotionCor2 [31] to produce drift-corrected images. Particles were picked and selected in RELION-2.1 [32], and contrast transfer function (CTF) parameters were estimated using CTFFIND4 [33] and integrated in RELION-2.1. Subsequent steps in three-dimensional (3D) reconstruction were carried out in RELION-2.1 in accordance with recommended gold-standard refinement procedures [32]. For all reconstructions, the final resolution was assessed using the standard FSC = 0.143 criterion. For FMDV-AWH-W2 complex, a total of 21999 particles were collected; 9406 particles were used for 3D reconstruction. For FMDV-AWH-W125 complex, a total of 12646 particles were collected; 7054 particles were used for 3D reconstruction (S5 Table).

### Model building and refinement

The X-ray structure of native FMDV O1BFS (PDB:1BBT) [34] was manually placed into the cryo-EM map for FMDV particles and rigid-body fitted with UCSF Chimera [35]. The X-ray

structure of native BOV-7 (PDB: 6E9U) [36] was manually placed onto the cryo-EM map for scFv and rigid-body fitted with UCSF Chimera [35]; fitting was further improved with real-space refinement using Phenix [37]. Manual model building was performed using Coot [38] in combination with real-space refinement with Phenix [37] to adjust mismatches between the model and the target protein. The density maps were kept constant during the entire fitting process, and the atomic coordinates were subjected to refinement. Additional structures reported in this work were built and refined by using FMDV (A/WH/CHA/09) particles as a starting model and rigid-body fitted and refined. Validation was conducted using the Mol-Probity function integrated within Phenix.

## Selection of neutralization-escape mutants using bovine mAbs

Neutralization escape mutants were generated by consecutive passages of FMDV in BHK-21 cells under the selective pressure of neutralizing mAbs, following a previously reported protocol with minor modifications [39]. The representative FMDV strain (A/AF72, A/WH/CHA/09 or A/GDMM/2013) was utilized to select mutants against these mAbs. Briefly, 10-fold serial dilutions of FMDV in 50 μl were incubated with 50 μl of various concentrations of mAbs (ranging from 20 μg/ml to 50 μg/ml) in 96-well microplates. Subsequently, the mixtures were used to infect BHK-21 cells ($10^6$ cells/ml) in a volume of 100 μl and incubated at 37°C for 48 h to allow virus propagation. First-passage viruses were harvested from wells seeded with the highest dilution of virus that produced an approximately 80 to 100% cytopathic effect (CPE). Further rounds of pressure selection were performed in 24-well plates, in which the passaged virus (200 μl) was incubated with an equal volume of a 2-fold concentration of antibodies in each well containing BHK-21 cells (400 μl). The harvested virus was subjected to several additional rounds of selection until it completely escaped neutralization after the addition of mAbs at concentrations of at least 400 μg/ml. The P1 region sequence corresponding to the obtained neutralization escape mutants was amplified by one-step reverse transcription-PCR (RT-PCR), as described previously [40], employing the primer pair Pan2041 (ACCTCCAACGGGTGGTACGC)/NK61 (GACATGTCCTCTTGCATCTG) and subsequently verified by sequencing. Mutated amino acids were determined by aligning the entire mutant P1 region with its initial parent virus sequence.

## Rescue of site-directed FMDV mutants by reverse genetics

Generation of full-length cDNAs was achieved using an existing pQQN plasmid harboring the entire P1 gene of A/WH/CHA/09. Site-directed mutagenesis was subsequently employed to introduce nucleic acid mutations, resulting in the production of full-length cDNAs with single amino-acid substitutions [41]. All mutant constructs were validated through nucleotide sequencing. The site-directed FMDV mutant viruses were rescued as previously reported [42]. Briefly, *Not*I-linearized mutant plasmids were transfected into BSR/T7 cells, following the manufacturer's instructions, using Lipofectamine 2000. The transfected cells were monitored daily for appearance of CPE. At 72 h post-transfection, the cells were harvested and passaged in BHK-21 cells. After 3 rounds of passaging, the mutant virus titers were determined in BHK cells by calculating the 50% tissue culture infectious dose ($TCID_{50}$), which was subsequently used to perform micro-neutralization assay as described above.

## Supporting information

**S1 Fig. Negative strain analysis of the purified FMDV 146S antigens before and after biotinylation.** Negative strain EM analysis of purified A/GDMM/2013 146S (A) and the resulting biotinylated A/GDMM/2013 (B). Negative strain EM analysis of purified A/AF72 146S (C)

and the resulting biotinylated A/AF72 (D).
(TIF)

**S2 Fig. Evaluation of reactivity of bovine mAbs with the 146S and 12S particles of A/AF72 strain (A) and A/GDMM/2013 strain (B) using indirect ELISA.** 12S particles were prepared from naïve 146S particles by acidification (incubation with $NaH_2PO_4$ (pH = 5.5) for 10 mins) or heat treatment for 1 h at 56°C. In indirect ELISA experiments, 100 ng/well of naïve 146S / acid treated 146S / heat treated 146S was respectively coated in 96-well plates overnight at room temperature. The plates were then washed three times with PBST (PBS buffer plus 0.05% Tween) and blocked with 1% gelatin in PBS at 37°C for 2 h. After three washes, the bovine mAbs at different concentrations were added and incubated at 37°C for 1 h. The plates were washed three times with PBST, and then the HRP-conjugated anti-His tag antibody (Genscript, China) at a dilution of 1:5,000 was added to the wells. The plates were then incubated at 37°C for 30 min and washed three times with PBST. Color was developed by adding 50 μl of TMB substrate (Pierce, Life Technology) for 10 min at room temperature. The process was stopped by adding equal volumes of 1 M $H_2SO_4$. Optical density at 450 nm ($OD_{450}$) was measured on a microplate reader (BioRad). The results represent one of three independent assays with duplication.
(TIF)

**S3 Fig. Cryo-EM analysis of FMDV-AWH-W2 complex (A) and FMDV-AWH-W125 complex (B).** Typical electron micrographs were collected with a defocus of 1.9 μm (FMDV-AWH-W2), 1.7 μm (FMDV-AWH-W125) (Scale bar, 1000 Å). Selected 2D class averages both show prominent spikes on the outer surface of viral particles (Scale bar, 480 Å). Fourier shell correlation (FSC) of the final 3D reconstruction after gold-standard refinement using RELION and THUNDER. The resolution corresponding to an FSC of 0.143 is shown for these virus-antibody complexes. FSC curves are plotted before (gray) and after (yellow) masking in addition to post-correction (orange), accounting for the effect of the mask using phase randomization.
(TIF)

**S4 Fig. Density maps of FMDV-AWH-W125 complex and FMDV-AWH-W2 complex.** Surface representation of the density maps for a protomer of FMDV-AWH-W125 complex (A) and FMDV-AWH-W2 complex (B). VP1, VP2, VP3 and VP4 of the protomer are blue, green, red and yellow; VH and VL of W125 are purple and orange, respectively; VH and VL of W2 are cyan and magenta, respectively. In the right panel, atomic models shown as sticks are superimposed to indicate the representative regions in wire frames. In the stick models, the residue numbers are indicated. The VP2, VP3, VH and VL residues are labeled with a subscript.
(TIF)

**S5 Fig. Analysis of conservation of key antigenic determinants on VP1, VP2 and VP3 of FMDV serotype A.** The full amino acids sequences of VP1, VP2 and VP3 of FMDV serotype A were downloaded from national center for biotechnology information (NCBI) as of June 30, 2023. The key antigenic determinants involved in common residues of A/WH/CHA/09, A/GDMM/2013 and A/AF72 were framed with rectangles and the conservation of corresponding residue was marked with orange. The key residues 58 (Q) and 147 (G, corresponding to RGD +2 position) on VP1 were determined by bnAbs W151, W153 and W145 (A). The key residues VP2 70 (T), 77 (H), 196 (Q) and VP3 61 (K) were targeted by bnAb W125 (B). The key residues 59 (D), 71 (Q), 76 (K), 84 (K) and 132 (T) on VP3 were targeted by bnAb W2 (C).
(TIF)

**S6 Fig. Binding modes of FMDV integrin receptor and antibody.** Binding modes of integrin (avβ6) receptor with scFv antibody W2 (A) and W125 (B). The panel shows a view down onto the capsid surface. VP1, VP2, VP3 and VP4 of the protomer are blue, green, red and yellow, respectively. The integrin and antibodies (W2 and W125) are drawn in cartoon representation; integrin is purple; W2 and W125 are respectively colored with orange and cyan. Black dashed circles show significant clashes between antibody (W2 and W125) and integrin receptor. (TIF)

**S1 Table. FMDV-AWH-W125 interaction residues.**
(DOCX)

**S2 Table. Tissue culture infective dose 50% (TCID$_{50}$) of the rescued mutant viruses.**
(DOCX)

**S3 Table. FMDV-AWH-W2 interaction residues.**
(DOCX)

**S4 Table. Strain-specific bovine neutralizing mAb escape mutants.**
(DOCX)

**S5 Table. Cryo-EM data collection and refinement statistics.**
(DOCX)

## Acknowledgments

We thank the Computing and Cryo-EM Platforms of Tsinghua University, Branch of the National Center for Protein Sciences (Beijing) for providing facilities. We thank the staff Shuyun Qi at Instrument Centre, Lanzhou Veterinary Research Institute, Chinese Academy of Agricultural Sciences for excellent assistance in single cell sorting using BD FACS Aria II.

## Author Contributions

**Conceptualization:** Kun Li, Yong He, Zhiyong Lou, Yimei Cao, Zengjun Lu.

**Data curation:** Kun Li, Yong He, Huifang Bao.

**Formal analysis:** Kun Li, Yong He, Li Wang, Pinghua Li, Ying Li.

**Funding acquisition:** Kun Li, Yong He, Yimei Cao, Zengjun Lu.

**Investigation:** Kun Li, Yong He, Li Wang, Shulun Huang, Shasha Zhou, Guoqiang Zhu, Yali Song, Ying Li, Sheng Wang, Qianliang Zhang.

**Methodology:** Kun Li, Yong He, Li Wang, Pinghua Li, Sheng Wang, Yimei Cao.

**Project administration:** Zhiyong Lou, Yimei Cao, Zengjun Lu, Zaixin Liu.

**Resources:** Kun Li, Yong He, Pinghua Li, Huifang Bao, Shasha Zhou, Pu Sun.

**Software:** Kun Li, Yong He, Shulun Huang, Xingwen Bai.

**Supervision:** Zhiyong Lou, Zengjun Lu, Zaixin Liu.

**Validation:** Kun Li, Li Wang, Huifang Bao, Yali Song, Zhixun Zhao.

**Visualization:** Kun Li, Yong He, Pinghua Li, Huifang Bao, Guoqiang Zhu, Pu Sun.

**Writing – original draft:** Kun Li, Yong He, Yimei Cao.

**Writing – review & editing:** Kun Li, Yong He, Yimei Cao, Zengjun Lu.

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
