## [Decision Letter · Decision Letter 0]

10 Jul 2023

Dear Dr. Lu,

Thank you very much for submitting your manuscript "Conserved antigen structures and antibody-driven variations on foot-and-mouth disease virus serotype A revealed by bovine neutralizing monoclonal antibodies" for consideration at PLOS Pathogens. As with all papers reviewed by the journal, your manuscript was reviewed by members of the editorial board and by several independent reviewers. In light of the reviews (below this email), we would like to invite the resubmission of a significantly-revised version that takes into account the reviewers' comments.

We apologize for the delay in obtaining reviews and reaching a decision. As you see, the work may be publishable, but not in its current form.

We cannot make any decision about publication until we have seen the revised manuscript and your response to the reviewers' comments. Your revised manuscript is also likely to be sent to reviewers for further evaluation.

Sincerely,

Vaughn Smider

Guest Editor

PLOS Pathogens

Sonja Best

Section Editor

PLOS Pathogens

Kasturi Haldar

Editor-in-Chief

PLOS Pathogens

orcid.org/0000-0001-5065-158X

Michael Malim

Editor-in-Chief

PLOS Pathogens

orcid.org/0000-0002-7699-2064

We apologize for the delay in obtaining reviews and reaching a decision. As you see, the work may be publishable, but not in its current form.

Reviewer's Responses to Questions

**Part I - Summary**

Reviewer #1: This study contains a lot of well-presented data, interesting information and proof-of-principle for how antigenic information in FMDV could potentially be used to engineer antigens capable of inducing more broadly reactive responses. If anything the significance of this is underplayed. There is scope to improve presentation of some of the rationale (suggestions elsewhere in this review).

Cattle were vaccinated with a trivalent O serotype vaccine and then sequentially boosted with three separate A serotype vaccines A/WH/CHA/09, A/GDMM/CHA/2013 and A/AF72. A/AF72 is a long-established vaccine strain related to A22, A/WH/CHA/09 is a vaccine strain of the Sea97 lineage and A/GDMM/CHA/2013 is a more recent strain also from the Sea97 lineage. The introduction explains these strains but nowhere in the abstract/summary or intro is it made clear that these are being used to immunize the cattle. Presumably these strains represent wide antigenic diversity within serotype A viruses and sequential immunization with these was hoped to stimulate cross reactive responses against conserved epitopes (this rationale was not immediately clear).

Then A/AF72 and A/GDMM/2013 were used as antigenically diverse serotype A antigens to select reactive mAbs from vaccinated cattle. The rationale here was clear from the start but how are these viruses antigenically diverse, is it possible to explain this somewhere? Were these two antigens used because they were the most divergent (e.g. in contrast A/WH/CHA/09 was considered to be in the middle?)?

B cells / mAbs isolated by reactivity with the older A/AF72 had better cross-reactive neutralization for the other (more recent) A strains, relative to the mAbs identified by reactivity with A/GDMM/2013.

For two mAbs isolated by reactivity for A/AF72 (but with cross reactivity with the other viruses) the structures of Fab binding to A/WH/CHA/09 was determined. Capsid amino acids in contact with the Fabs were mutated and tested for effect on neutralization by the mAbs. The residues required for recognition and neutralization by both mAbs were conserved across all 3 type A viruses used in the study.

Further residues contributing to cross reactive neutralization were identified by immune escape from treatment with additional cattle mAbs.

Immune escape using panels of strain specific neutralizing mAbs were used to show how an epitope in the earlier A serotype viruses was no longer present in the newer serotype virus and conversely how a new epitope had arisen which was not present in the earlier viruses. The frequency of immune escape (flexibility) at different positions in the capsid of the different viruses suggested that different strains or lineages within a serotype may have different patterns of evolution.

Reviewer #2: In this study by Li et al, the authors aimed to understand the structures of conserved and variable antigenic sites on foot-and-mouth disease virus (FMDV) particles. The work specifically investigates serotype A FMDV, generally considered to be one of the most antigenically diverse serotypes, however, the sites/structures of all variable and conserved antigenic sites are not fully resolved. Using two serotype A strains (one historical and one current) as bait antigen the authors identify and characterise 39 strain specific and 5 broadly neutralising antibodies. Using cryo-EM two conserved antigenic sites were proposed that involve a complex of residues on viral proteins VP1, VP2 and VP33. In contrast, stain specific neutralising antibodies involved more key residues in VP3 that were more variable between strains. The authors proposes VP3 and VP1 are more flexible with VP2 being more structurally rigid, which could drive greater broadly neutralising antibody production.

Foot-and-mouth disease virus is an economically important pathogen. Vaccines are likely key for future global control of infection and therefore understanding antigenicity is a relevant area of study. The authors present a commendable amount of data that convincingly characterised specific and more broadly neutralising antigenic sites of FMDV capsids. This work therefore incrementally advances the knowledge on FMDV antigenicity. With some exceptions (as described below) the data is largely presented well and is well interpreted but it does not support the overall conclusion on viral structural flexibility and the manuscript contains multiple wording, spelling or grammatical errors. Specific points to improve the study are listed below:

**Part II – Major Issues: Key Experiments Required for Acceptance**

Reviewer #1: N/A

Reviewer #2: 1. Line 391: The authors use 146S antigen for immunisation. The production/purification of the antigen was not described. How can you ensure it is not contaminated with other FMDV particles that would be immunogenic (e.g., 45S particles).

2. Following on from point 1, do the bovine monoclonal antibodies identified recognise a 75S, 45S particle or 12S subunit, or do these particles generate the same antibodies. Understanding this is very important to understand if non-infectious particles/VLPs/antigens could illicit an appropriate immune response to be used as a vaccine candidate.

3. Figure 3E/4E showing individual single IF images does not show recovery of mutant viruses convincingly, in particular since such a high MOI (10) was used for these experiments. A titre of the recovered mutant viruses would be more convincing and inform on fitness costs of these mutations. Understanding the fitness of these mutant viruses is essential for understanding the antibody neutralisation results.

4. Lines 271-274: I agree the author’s data suggests that VP3 is more readily able to change sequence under immune pressure, however the data presented does not investigate structural flexibility. The authors study does not look at structural dynamics/movements of VP2 or VP3/1 but sequence tolerance/variability and ability of these proteins to sample sequence space under immune pressure. The authors should modify the wording of their conclusions here and elsewhere in the discussion.

**Part III – Minor Issues: Editorial and Data Presentation Modifications**

Reviewer #1: Abstract - Why not include mention of A22 and SEA97 lineages in the abstract? Why not make clear how the animals have been immunized ?

The introduction is pretty short.

After reading abstract, author summary (and introduction) I still did not understand what the cattle were immunised with. This information only seems to be provided in the methods. Give that so much emphasis is placed on describing the diverse bait antigens it feels strange that no rationale is given for the immunisation strategy. If the objective of the study was to isolate mAbs which were broadly cross reactive within the A serotype, why were the cattle first immunised with O serotype vaccines? Was this a single trivalent O vaccine? Then three A serotype vaccines are given as separate sequential boosters. Please explain the rationale for the immunisation in the results and at least mention something in the abstract. Could the initial immunisation with O serotype vaccines also influence the response to the later A vaccines due to ‘original antigenic sin’ which may constrain the direction of responses? Perhaps worth a mention in the discussion?

Was any other data generated to confirm that the immunisations had worked as expected? ELISA, VNT on resulting sera?

For the selection of mAbs recognising conserved epitopes would it have been useful to attempt to find B cells sorted by labelling with both A/AF72 and A/GDMM/2013 antigens?

The Fab-virus structures are generated with A/WH/CHA/09, it would be useful to explain why this was done when none of the B cells were selected against this antigen.

Specific comments on language use.

22 “Ancient”: I’m not sure what this is supposed to mean but I don’t think a virus strain from 1972 qualifies as ancient. Is this the best phrase to use?

28 “sole VH of W2”: what does this mean? There was no mention of this detail for W125 (line 26) so is the situation with W2 something special or unusual?

30-32 Not sure if I understand this section but I think “The highly conserved epitopes…” would make more sense if it read “Additional highly conserved epitopes…”

84 Statement “In China, there are three representative strains of FMDV serotype A…”. Please clarify what this means. The sentence at the end of the paragraph suggests these are intended to represent antigenic diversity or antigenic evolutionary history of A serotype viruses in China… it would be useful to indicate this at the start of the paragraph.

137 I don’t think W125 and W2 have been introduced in the main text yet.

152/153 what do “CCP4” and “RIVEM” stand for? LCDR?

387 “…basic immunization with the three topotypes of FMDV serotype O…” what does basic mean? Was this a single trivalent vaccine or separate sequential vaccines?

Various other examples where wording or language could be improved. Please check.

Reviewer #2: 1. In several places (e.g., figures 3G, 4G and lines 173-174) the authors state amino acids are “highly conserved” but only align the three sequences used in this study. Making reference to figure 5 where a larger alignment is conducted would offer more insight on the conservation of the residues investigated.

2. I feel the abstract contains too much detail/assumed information (e.g., long list of amino acid interactions and names of viral proteins). To make it accessible to the broader readership I would consider changing the abstract to make it more accessible to non-FMDV experts.

3. Lines 142-143: The authors conclude a total of 60 scFv were bound to each virion. Using icosahedral averaging for the cryo-EM reconstructions can the authors be sure of this occupancy?

4. Lines 20-21: please include additional references to support the antigenic diversity of serotype A FMDV and previous work on FMDV serotype A antigenicity such as PMID: 36699337.

5. Lines 54-55: there have been no reports of serotype C FMDV since 2004 and it is now considered to be extinct. Please modifying the wording.

6. Lines 71-72: please include references for the identification of antigenic sites based on murine antibodies.

7. Lines 111-112: “the proportions of A/AF72-binding B cells was 0.06%”. I cannot see how this number is calculated from tables 1F or 1G.

8. Lines 120: define VNT

9. Figure 1: some panels of data could be relocated to supplementary to streamline the figure, such as panels A-C.

10. Figure 3A has become distorted (i.e., out of aspect ratio) and is therefore hard to interpret.

11. Line 158: should refer to figure 3D not 2D.

12. Lines 169-204 in references to figures 3/4 and lines 261-296 in reference to figure 7: it is confusing to refer to the panels of data out of order. i.e., the text refers to figure 7E before 7A. It would be easier to follow if the panels of data are in the same order as they are referred to in the text.

13. Lines 449-456: how many particles were collected in the collection and used for the cryo-EM reconstructions.

PLOS authors have the option to publish the peer review history of their article (what does this mean?). If published, this will include your full peer review and any attached files.

Reviewer #1: No

Reviewer #2: No
---

## [Decision Letter · Decision Letter 1]

25 Oct 2023

Dear Dr. Lu,

Thank you very much for submitting your manuscript "Conserved antigen structures and antibody-driven variations on foot-and-mouth disease virus serotype A revealed by bovine neutralizing monoclonal antibodies" for consideration at PLOS Pathogens. As with all papers reviewed by the journal, your manuscript was reviewed by members of the editorial board and by several independent reviewers. The reviewers appreciated the attention to an important topic. Based on the reviews, we are likely to accept this manuscript for publication, providing that you modify the manuscript according to the review recommendations.

Again we apologize for delays in review of your manuscript. We intend to accept the manuscript upon receipt of the minor revisions requested by Reviewer 1 below, which we believe will improve the manuscript. Please provide the updated manuscript and we will endeavor to proceed expeditiously. Thank you.

Sincerely,

Vaughn Smider

Guest Editor

PLOS Pathogens

Sonja Best

Section Editor

PLOS Pathogens

Kasturi Haldar

Editor-in-Chief

PLOS Pathogens

orcid.org/0000-0001-5065-158X

Michael Malim

Editor-in-Chief

PLOS Pathogens

orcid.org/0000-0002-7699-2064

Hello,

Again we apologize for delays in review of your manuscript. We intend to accept the manuscript upon receipt of the minor revisions requested by Reviewer 1 below, which we believe will improve the manuscript. Please provide the updated manuscript and we will endeavor to proceed expeditiously. Thank you.

Reviewer Comments (if any, and for reference):

Reviewer's Responses to Questions

**Part I - Summary**

Reviewer #1: Thanks to the authors for responding to my questions. Please add the explanations you have given into the manuscript as suggested later (Part III - Minor Issues).

Reviewer #2: The authors have satisfactorily addressed my concerns.

**Part II – Major Issues: Key Experiments Required for Acceptance**

Reviewer #1: (No Response)

Reviewer #2: (No Response)

**Part III – Minor Issues: Editorial and Data Presentation Modifications**

Reviewer #1: Response to Q2. Regarding the potential effect of previous sequential immunisation with O serotype antigens, nothing was changed significantly in the revised manuscript. It is still only mentioned in the methods. The potential effect of this on the responses in the current study should be acknowledged – it would be easy to mention this in the discussion, please add something.

Response Q3. Thank you for the reply (after final immunisation sera could neutralise all three strains). Why does this indicate that each sequential immunisation has been effective? For example, how can you rule out that only one of the three immunisations were effective and that the sera simply cross reacts to neutralise all three strains? I strongly recommend you add information about this or acknowledge it in the discussion.

Response to Q5. Thank you this is useful. Please can this this explanation be added to the manuscript?

Q11. For modified text, suggest: “Briefly, bovine had been previously sequentially immunized with the three separate topotypes…”

Reviewer #2: (No Response)

PLOS authors have the option to publish the peer review history of their article (what does this mean?). If published, this will include your full peer review and any attached files.

Reviewer #1: No

Reviewer #2: No

Figure Files:

Data Requirements:

Reproducibility:

References:

---

## [Editor Report · Decision Letter 2]

7 Nov 2023

Dear Dr. Lu,

We are pleased to inform you that your manuscript 'Conserved antigen structures and antibody-driven variations on foot-and-mouth disease virus serotype A revealed by bovine neutralizing monoclonal antibodies' has been provisionally accepted for publication in PLOS Pathogens.

Best regards,

Vaughn Smider

Guest Editor

PLOS Pathogens

Sonja Best

Section Editor

PLOS Pathogens

Kasturi Haldar

Editor-in-Chief

PLOS Pathogens

orcid.org/0000-0001-5065-158X

Michael Malim

Editor-in-Chief

PLOS Pathogens

orcid.org/0000-0002-7699-2064
---

## [Editor Report · Acceptance letter]

15 Nov 2023

Dear Dr. Lu,

We are delighted to inform you that your manuscript, "Conserved antigen structures and antibody-driven variations on foot-and-mouth disease virus serotype A revealed by bovine neutralizing monoclonal antibodies," has been formally accepted for publication in PLOS Pathogens.

Best regards,

Kasturi Haldar

Editor-in-Chief

PLOS Pathogens

orcid.org/0000-0001-5065-158X

Michael Malim

Editor-in-Chief

PLOS Pathogens

orcid.org/0000-0002-7699-2064